# Strengthen Out-of-Distribution Detection Capability with Progressive Self-Knowledge Distillation

**Yang Yang** [1]   **Haonan Xu** [1]

## Abstract

Out-of-distribution (OOD) detection aims to ensure AI system reliability by rejecting inputs outside the training distribution. Recent work shows that memorizing atypical samples during later stages of training can hurt OOD detection, while strategies for forgetting them show promising improvements. However, directly forgetting atypical samples sacrifices ID generalization and limits the model's OOD detection capability. To address this issue, we propose Progressive Self-Knowledge Distillation (PSKD) framework, which strengthens the OOD detection capability by leveraging self-provided uncertainty-embedded targets. Specifically, PSKD adaptively selects a self-teacher model from the training history using pseudo-outliers, facilitating the learning of uncertainty knowledge via multi-level distillation applied to features and responses. As a result, PSKD achieves better ID generalization and uncertainty estimation for OOD detection. Moreover, PSKD is orthogonal to most existing methods and can be integrated as a plugin to collaborate with them. Experimental results from multiple OOD scenarios verify the effectiveness and general applicability of PSKD.

## 1. Introduction

In real-world scenarios, deep learning models are inevitably exposed to unseen classes of samples that lie outside the training distribution (Yang et al., 2021), known as OOD data. Persisting in classifying OOD data into one of the predefined training classes is meaningless. Such unreliable behavior could lead to disastrous consequences, particularly in safety-critical fields like autonomous driving (Geiger et al., 2012) and medical diagnosis (Litjens et al., 2017).

[1]Nanjing University of Science and Technology. Correspondence to: Yang Yang <yyang@njust.edu.cn>.

*Proceedings of the 42nd International Conference on Machine Learning*, Vancouver, Canada. PMLR 267, 2025. Copyright 2025 by the author(s).

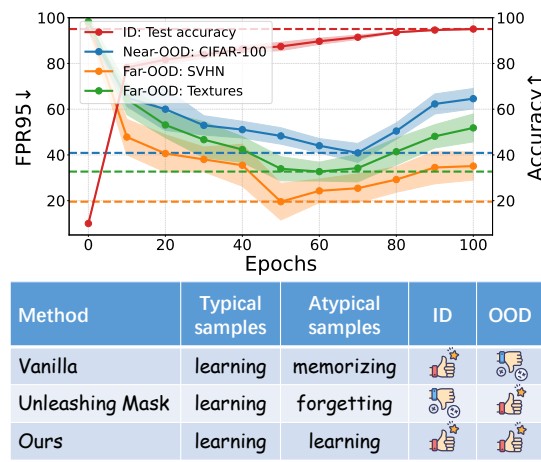

Figure 1: **Upper:** The FPR95 (lower values indicate better OOD detection) curves for various OOD datasets and the ID test accuracy curve, tracked when CIFAR-10 is used as the ID dataset for training. **Lower:** Comparisons of existing methods and our PSKD for handling atypical samples.

This highlights the necessity of OOD detection to ensure AI system reliability by enabling models to identify and reject predictions for OOD inputs.

Up to now, many efforts have been made in pursuing reliable detection methods. Early methods, represented by MSP (Hendrycks & Gimpel, 2017), ODIN (Liang et al., 2018) and Energy (Liu et al., 2020), focus on developing suitable scoring functions for OOD uncertainty estimation. However, having a well-trained model with a solid basis for OOD detection is an essential prerequisite. To this end, a series of studies delve into training-time regularization methods, which can be categorized into two main areas (Zhang et al., 2023b). The first involves exploring alternative training strategies to train stronger models with better discriminative representations (DeVries & Taylor, 2018; Sehwag et al., 2021; Wei et al., 2022; Pei, 2024). The second involves introducing realistic outliers to help OOD detectors learn more robust decision boundaries (Hendrycks et al., 2019; Liu et al., 2020; Wang et al., 2023; Jiang et al., 2024). Despite the promising performance achieved by these methods, limited attention is given to the models' intrinsic OOD detection capabilities, which constrains their full potential.

To provide an intuitive explanation of this issue, we monitor the model's performance on the ID classification and OOD detection throughout the entire training process, as shown in Figure 1. The results show that the model's optimal OOD detection performance occurs at an intermediate stage before convergence, rather than at the final well-trained state. Moreover, Zhu et al. (2023) have confirmed that this phenomenon is prevalent across a variety of training settings. They attribute this to the model's tendency to memorize atypical samples during later stages of training, which benefits ID generalization but leads the model to become more confident about OOD data, thereby hurting OOD detection (Zhu et al., 2023). In response, they propose Unleashing Mask / Unleashing Mask Adopts Pruning (UM/UMAP), which uses gradient ascent to forget these atypical samples, thereby backtracking the model to an earlier state with better OOD detection capabilities. However, determining the extent of model backtracking is labor-intensive and needs to be empirically found by the users. More importantly, directly forgetting atypical samples can sacrifice ID generalization and limit the model's OOD detection capability.

Targeting these important problems, we propose a novel self-learning framework termed Progressive Self-Knowledge Distillation (PSKD), designed to leverage self-provided uncertainty-embedded targets for more effective learning from both typical and atypical samples. Concretely, PSKD adaptively selects the teacher model with the most reliable uncertainty estimation from the current training history, which can be assisted by generated pseudo-outliers. Multi-level distillation, applied to both features and responses, is then used to facilitate more effective learning of OOD knowledge from uncertainty-embedded targets provided by the teacher model. This can mitigate overconfident predictions caused by treating training samples with varying uncertainties uniformly via one-hot targets. As a result, PSKD achieves more reliable uncertainty estimation for OOD detection, while also offering the additional benefit of improved ID generalization. Moreover, PSKD can be integrated as a plugin, leveraging self-distillation to restore the model's intrinsic OOD detection capability. This establishes a solid OOD detection model basic for most existing methods, thereby pushing their performance further.

## 2. Related Work

### 2.1. Out-of-Distribution Detection

**OOD Scoring Methods** aim to provide suitable measures to indicate the likelihood that a sample originates from the OOD distribution (Lee et al., 2018; Huang et al., 2021; Djurisic et al., 2023; Xu et al., 2024; Xu & Yang, 2025). For example, MSP (Hendrycks & Gimpel, 2017) directly uses the maximum SoftMax score as the criterion. ODIN (Liang et al., 2018) improves the MSP score by introducing

input perturbations and temperature scaling. Energy score (Liu et al., 2020) utilizes the logsumexp of the output logits, which is provably aligned with the density of inputs. Our work is orthogonal to these methods, as it focuses on establishing a stronger model foundation for OOD detection.

**Training-Time Regularization Methods** aim to calibrate the model to address OOD detection. One major category of methods involves altering training strategies to provide better discriminative representations for OOD detection (Zhou et al., 2021; Huang & Li, 2021; Zhang & Xiang, 2023). LogitNorm (Wei et al., 2022) and T2Fnorm (Regmi et al., 2024) provide alternatives to cross-entropy loss by decoupling the impact of logits and feature norms, respectively. SNN (Ghosal et al., 2024) learns the most relevant subspace to tackle the curse-of-dimensionality problem in distance-based OOD detection. Zhu et al. (2023) propose UM/UMAP to revert a well-trained model to an earlier stage, aiming to forget memorized atypical samples which hurt OOD detection. Another category introduces realistic outliers to facilitate the model in learning more robust decision boundaries (Li & Vasconcelos, 2020; Ming et al., 2022; Yang et al., 2023; 2024). OE (Hendrycks et al., 2019) forces the predictive distribution of outliers to be uniform. MixOE (Zhang et al., 2023a) mixes ID data and outliers to expand the coverage of different OOD granularities. DOE (Wang et al., 2023) utilizes model perturbations to implicitly broaden the coverage of outliers for better generalization. While effective in OOD detection, these methods often degrade ID task performance, and realistic outliers are typically costly to acquire.

### 2.2. Knowledge Distillation

Knowledge distillation is a popular learning method where a student model is trained to mimic the behavior of a more powerful teacher model (Gou et al., 2020; Cao et al., 2023). The idea originally stems from Bucila et al. (2006), and is later formally popularized by Hinton et al. (2015). Different knowledge can be utilized, represented by response-based knowledge (Hinton et al., 2015) and feature-based knowledge (Romero et al., 2015; Ahn et al., 2019). Self-distillation is a special case of knowledge distillation, where the same network is adopted for both the teacher and student models. Intuitively, it enables the model to refine itself via extended learning of its self-knowledge. In this context, Zhang et al. (2019) explore distilling knowledge from the deeper layers of the network into its shallower layers. Yang et al. (2019) propose transferring knowledge from the early stages (teacher) of the network to the later stages (student) to facilitate the supervised training process. Moreover, Mobahi et al. (2020) provide a theoretical analysis of self-distillation, and Zhang & Sabuncu (2020) empirically demonstrate its improved performance.

## 3. Preliminaries

**Setup.** In this paper, we focus on the setting of $K$-way image classification. Formally, let $\mathcal{X}$ be the input space and $\mathcal{Y}_{\text{in}} = \{1, 2, ..., K\}$ be the ID label space. The learner has access to a labeled training set $\mathcal{D}_{\text{train}} = \{(\mathbf{x}_i, \mathbf{y}_i)\}_{i=1}^n$ and a validation set $\mathcal{D}_{\text{val}}$, where the samples are drawn *i.i.d.* from a joint distribution $\mathcal{P}_{\mathcal{X}\mathcal{Y}_{\text{in}}}$. Let $f_\theta : \mathcal{X} \to \mathbb{R}^K$ denote the classification model parameterized by $\theta$, which is learned by minimizing the empirical risk: $\theta^* = \arg\min_\theta R_{\text{ID}}(f_\theta)$, where $R_{\text{ID}}(f_\theta) = \mathbb{E}_{(\mathbf{x},\mathbf{y}) \in \mathcal{D}_{\text{train}}} \ell(f_\theta(\mathbf{x}), \mathbf{y})$ and $\ell$ is the loss function. Generally, a SoftMax layer with temperature $T$ (normally set to 1) is applied for the prediction $f_\theta(\mathbf{x})$:

$$p_i = \frac{\exp(z_i/T)}{\sum_j \exp(z_j/T)}, \tag{1}$$

where $z_i$ is the logit for the $i$-th class. A larger $T$ can make the probability distribution softer, which can be beneficial for knowledge distillation (Hinton et al., 2015).

**OOD Detection** aims to discern OOD data arising from an irrelevant distribution whose label set has no intersection with $\mathcal{Y}_{\text{in}}$ (Yang et al., 2021). In general, the OOD detector $g_\tau(\cdot)$ is given by:

$$g_\tau(\mathbf{x}) = \begin{cases} \text{ID}, & \text{if } S(\mathbf{x}; f_\theta) > \tau; \\ \text{OOD}, & \text{if } S(\mathbf{x}; f_\theta) \leq \tau, \end{cases} \tag{2}$$

where $\tau$ is a threshold typically chosen to correctly classify the majority of ID data (*e.g.*, 95%), and $S(\mathbf{x}; f_\theta) : \mathcal{X} \to \mathbb{R}$ is the OOD scoring function, which quantifies model uncertainty as a scalar value. Representative scoring functions include the MSP (Hendrycks & Gimpel, 2017) and Energy score (Liu et al., 2020), which are defined as follows:

$$S_{\text{MSP}}(\mathbf{x}; f_\theta) = \max_i p_i, \; S_{\text{Energy}}(\mathbf{x}; f_\theta) = \log \sum_i \exp(z_i). \tag{3}$$

Since OOD labels lie outside the training label space, the model tends to produce lower expected scores compared to ID cases. However, OOD detection remains non-trivial because deep models can be overconfident when faced with OOD data (Nguyen et al., 2015; Bendale & Boult, 2016).

## 4. Proposed Method

### 4.1. Motivation and Framework

As illustrated in Figure 1, the model's optimal OOD detection performance is achieved at an intermediate stage of training before full convergence, rather than at the final well-trained state. In other words, a trade-off exists between pursuing better performance of the main task (ID prediction) and the OOD detection task. Zhu et al. (2023) attribute this inconsistency to the model's tendency to memorize atypical

samples, which improves ID generalization but simultaneously makes the model more confident about OOD data, thereby hindering OOD detection. Accordingly, Zhu et al. (2023) propose UM and its variant UMAP, which directly forget atypical samples to revert the model to an earlier state. Conversely, in this paper, we explore how to learn from valuable atypical samples effectively.

Typically, the training set in supervised learning consists of pairs of an input image and its corresponding one-hot target. However, in the context of uncertainty estimation for OOD detection, assigning all training samples an absolutely confident one-hot target can be a potential hindrance. This could hinder the model's ability to effectively learn to differentiate between samples with varying levels of uncertainty, potentially leading to overconfidence in its uncertainty estimates when encountering OOD data. Therefore, providing sample-level uncertainty knowledge as the learning objective for OOD detectors is crucial. Intuitively, atypical samples should be assigned higher uncertainty soft targets for learning compared to typical samples. However, a practical challenge is that hand-crafting uncertainty-embedded targets is prohibitively costly, requiring substantial expert knowledge and labor-intensive efforts.

To this end, we propose a self-learning paradigm PSKD that leverages uncertainty-embedded targets, self-provided by the teacher model selected from the current training history, as a substitute to facilitate the model's learning of uncertainty knowledge. Figure 2 illustrates the framework of our proposal. Conceptually, PSKD jointly optimizes for both: (1) accurate ID classification, and (2) reliable uncertainty estimation. Formally, let the self-selected OOD teacher model be denoted as $f_{\theta^*}$, the overall risk can be formalized with a weight factor $\lambda$ as follows:

$$\arg\min_\theta \; [\underbrace{(1-\lambda) \cdot R_{\text{ID}}(f_\theta)}_{\text{ID classification}} + \underbrace{\lambda \cdot R_{\text{OOD}}(f_\theta; f_{\theta^*})}_{\text{Self OOD knowledge distillation}}], \tag{4}$$

where the term $R_{\text{ID}}(f_\theta)$ is intended to learn ID classification capability, which can be cross-entropy loss in vanilla training, while $R_{\text{OOD}}(f_\theta; f_{\theta^*})$ is designed to learn uncertainty estimation knowledge from the self-selected OOD teacher model. To prevent large bias in the teacher model's responses during early training, we dynamically adjust the weights of ID learning and self-distillation for effective initial learning. Mathematically, the weight factor $\lambda$ at epoch $e$ is computed using a cosine annealing schedule as follows:

$$\lambda = \frac{\lambda_{\text{final}}}{2}\left(1 - \cos\left(\frac{e}{E}\pi\right)\right), \tag{5}$$

where $\lambda_{\text{final}}$ is the final self-distillation weight, and $E$ is the total number of training epochs. To make PSKD feasible, we have two key challenges that need to be addressed: (1) how to identify and select a desired teacher model $f_{\theta^*}$, and

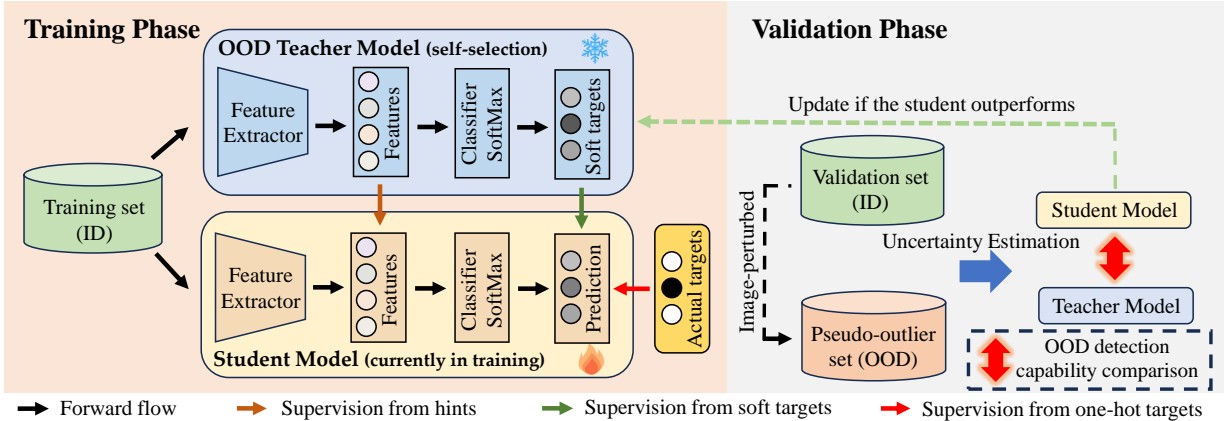

Figure 2: Illustration of model training in a self-learning paradigm with PSKD. During the training phase, the model is jointly supervised by both actual targets and uncertainty-embedded teacher signals. In the validation phase, pseudo-outliers are used with the validation set for OOD detection capability comparison. The OOD teacher model is replaced by the student model when the student performs OOD detection better. This process can be iteratively repeated over time.

(2) how to effectively learn OOD knowledge from the self-selected teacher model.

### 4.2. Self-Selection of the OOD Teacher Model

To acquire the OOD teacher model, the first crucial step is to define explicit criteria for assessing and comparing the OOD detection performance of different models. In this work, we adopt the AUROC (Area Under the Receiver Operating Characteristic curve), a widely recognized metric in OOD detection, as it provides a comprehensive evaluation across various decision thresholds. Given a model $f_\theta$, an ID dataset $\mathcal{D}_{\text{ID}}$, an OOD dataset $\mathcal{D}_{\text{OOD}}$, and the OOD scoring function $S(\cdot; f_\theta)$, the OOD detection capability of $f_\theta$ can be approximately quantified using the AUROC as follows:

$$A(f_\theta) = \frac{\sum_{\mathbf{x} \in \mathcal{D}_{\text{ID}}} \sum_{\mathbf{x}' \in \mathcal{D}_{\text{OOD}}} \mathbb{I}(S(\mathbf{x}; f_\theta) > S(\mathbf{x}'; f_\theta))}{|\mathcal{D}_{\text{ID}}| \cdot |\mathcal{D}_{\text{OOD}}|}, \quad (6)$$

where $\mathbb{I}(\cdot)$ denotes the indicator function and $|\mathcal{D}|$ denotes the cardinality of the set $\mathcal{D}$. The validation dataset $\mathcal{D}_{\text{val}}$ can naturally serve as a reference for $\mathcal{D}_{\text{ID}}$. As for $\mathcal{D}_{\text{OOD}}$, we adopt a generic outlier generation strategy by perturbing or corrupting images from $\mathcal{D}_{\text{val}}$ to create the OOD data, as obtaining realistic outliers is often costly or impractical. The generation details and the impact of using pseudo-outliers from different sources, as well as the choice of $S(\cdot; f_\theta)$, are further discussed in Section 5.4.

In the context of the evaluation criteria, a strong OOD detector is expected to assign higher scores to ID cases than OOD cases, resulting in a higher $A(\cdot)$. To maintain the advanced uncertainty estimation capability of the teacher, PSKD dynamically selects the teacher model that achieves the best $A(\cdot)$ determined from the current training history,

periodically after a certain number of training steps (*e.g.*, after one epoch in the validation stage).

### 4.3. Self-Knowledge Distillation

To efficiently learn OOD knowledge from the teacher model, we focus on knowledge at the response and penultimate feature levels, which are most relevant and impactful for OOD detection (Zhu et al., 2022). At the response level, PSKD leverages self-provided uncertainty-embedded targets derived from the teacher's predictions, enabling the model to learn uncertainty knowledge that reflects the teacher's reliable behavior in uncertain scenarios. This can be achieved by minimizing the Kullback-Leibler (KL) divergence between the output probabilities of the classifiers:

$$R_{\text{logits}}(f_\theta; f_{\theta^*}) = \mathbb{E}_{\mathbf{x} \in \mathcal{D}_{\text{train}}} \text{KL}\left(p_{f_\theta^*}(\mathbf{x}) \,\|\, p_{f_\theta}(\mathbf{x})\right), \quad (7)$$

where $p_{f_\theta^*}(\mathbf{x})$ and $p_{f_\theta}(\mathbf{x})$ represent the softmax outputs from the teacher and student models, respectively.

On the other hand, feature-level supervision comes from the deepest classifier's hint, defined as the teacher model's feature representation guiding the student's learning (Hinton et al., 2015). This is achieved by reducing the distance between feature maps in the teacher model and the student model:

$$R_{\text{feat}}(f_\theta; f_{\theta^*}) = \mathbb{E}_{\mathbf{x} \in \mathcal{D}_{\text{train}}} \|h_{\theta^*}(\mathbf{x}) - h_\theta(\mathbf{x})\|_2^2, \quad (8)$$

where $h_{\theta^*}(\mathbf{x})$ and $h_\theta(\mathbf{x})$ denote the penultimate feature representations of the teacher and student models, respectively.

To sum up, the self-OOD knowledge distillation risk can be formalized with a weighting factor $\alpha$ as follows:

$$R_{\text{OOD}}(f_\theta; f_{\theta^*}) = R_{\text{logits}}(f_\theta, f_{\theta^*}) + \alpha \cdot R_{\text{feat}}(f_\theta, f_{\theta^*}). \quad (9)$$

The pseudo code of PSKD is available in Appendix A.

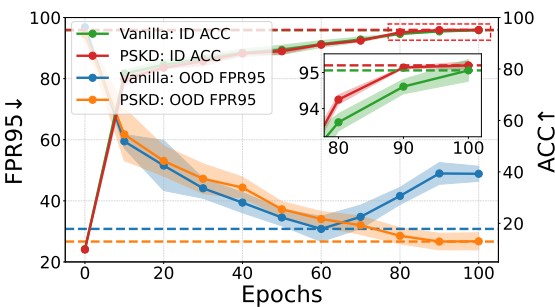

Figure 3: Performance tracking curves for ID prediction and OOD detection tasks. The OOD results are averaged over near- and far-OOD groups on the CIFAR-10 benchmark.

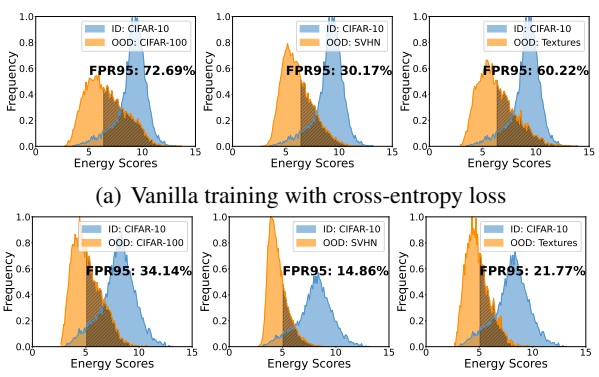

(a) Vanilla training with cross-entropy loss

(b) Training with our proposed PSKD

Figure 4: Changes in OOD score distribution using PSKD.

### 4.4. Insight Justification

The following remarks are provided to empirically elaborate on the insights of PSKD.

**Remark 1. PSKD restores the model's intrinsic OOD detection capability and also benefits ID generalization.** As shown in Figure 3, in the later stages of vanilla training, the model's OOD detection capability weakens due to its memorization of atypical samples. In this regard, PSKD restores the model's intrinsic OOD detection capability through self-distillation and further enhances it by effectively learning uncertainty from atypical samples. Moreover, PSKD achieves additional improvements in the original task by making better use of atypical samples, without requiring a trade-off between ID task performance and OOD detection performance, as would occur when directly forgetting valuable atypical samples.

**Remark 2. PSKD alleviates the overconfidence of OOD data by learning with uncertainty-embedded targets.** Traditional supervision is provided via one-hot vectors, which fails in uncertainty learning by assigning absolute confidence to samples with varying levels of uncertainty. This can make the model prone to overconfidence when confronted with OOD samples. To address this problem,

our proposed leverages the potential within the model's own learning process to generate soft targets that inherently capture sample-level uncertainty. Specifically, PSKD alleviates overfitting to atypical samples by learning from the uncertainty-embedded targets provided by the self-selected teacher model, thereby effectively enhancing the model's ability to perceive uncertainty. As shown in Figure 4, PSKD significantly reduces the model's overconfidence in OOD samples, as evidenced by the decrease in the right tail of the OOD score distribution for OOD data, thereby improving ID-OOD separation.

## 5. Experiments

In this section, we first describe our experimental setup, then present the main results on multiple OOD detection benchmarks, followed by ablation studies and further analysis.

### 5.1. Experimental Setup

Following standard practices (Zhang et al., 2023b), we evaluate our PSKD using the OpenOOD v1.5 benchmark on both small-scale CIFAR and large-scale ImageNet, covering near-OOD scenarios with semantic shifts and far-OOD scenarios with further obvious covariance shifts [1].

**Datasets.** For small-scale experimental setups, CIFAR-10/100 (Krizhevsky, 2009) is adopted as the ID dataset. The near-OOD group contains CIFAR-100/10 and Tiny-ImageNet (Le & Yang, 2015), while the far-OOD group consists of MNIST (Deng, 2012), SVHN (Netzer et al., 2011), Textures (Cimpoi et al., 2014), and Places365 (Zhou et al., 2018). For the realistic outlier dataset, we follow the OpenOOD standards and use TinyImageNet-597 (Le & Yang, 2015), which has no category overlap with CIFAR-10/100 and the OOD test sets.

Large-scale experiments are conducted on ImageNet-200, a 200-class subset of ImageNet-1K (Deng et al., 2009), as the ID dataset. Evaluations cover near-OOD scenarios, including SSB-hard (Vaze et al., 2022) and NINCO (Bitterwolf et al., 2023), as well as far-OOD scenarios, including iNaturalist (Horn et al., 2018), Textures (Cimpoi et al., 2014), and OpenImage-O (Wang et al., 2022). The rest 800 classes from ImageNet-1K are used as realistic outlier data.

**Evaluation Metrics.** We report three widely adopted metrics: (1) FPR95, the false positive rate of OOD data at a 95% true positive rate of ID data; (2) AUROC, commonly referred to as AUC, is used here following (Zhang et al., 2023b) to denote the area under the receiver operating characteristic curve; and (3) ID ACC, the classification accuracy on the ID test set.

---

[1]The code for this work is available at https://github.com/njustkmg/ICML25-PSKD

Table 1: Comparison on CIFAR benchmarks. All values are percentages, and OOD detection results are averaged over multiple OOD test datasets. The best results are in bold, with the second-best underlined. Detailed results for each OOD dataset are provided in Appendix G. ↑ indicates that larger values are better, while ↓ indicates that smaller values are better.

| Method | CIFAR-10 | | | | | CIFAR-100 | | | | |
|---|---|---|---|---|---|---|---|---|---|---|
| | Near-OOD | | Far-OOD | | ID ACC↑ | Near-OOD | | Far-OOD | | ID ACC↑ |
| | FPR95↓ | AUROC↑ | FPR95↓ | AUROC↑ | | FPR95↓ | AUROC↑ | FPR95↓ | AUROC↑ | |
| *OOD scoring methods (vanilla training with cross-entropy)* | | | | | | | | | | |
| MSP | $48.17_{\pm3.92}$ | $88.03_{\pm0.25}$ | $31.72_{\pm1.84}$ | $90.73_{\pm0.43}$ | $95.06_{\pm0.30}$ | $\mathbf{54.80_{\pm0.33}}$ | $80.27_{\pm0.11}$ | $58.70_{\pm1.06}$ | $77.76_{\pm0.44}$ | $77.25_{\pm0.10}$ |
| ODIN | $76.19_{\pm6.08}$ | $82.87_{\pm1.85}$ | $57.62_{\pm4.24}$ | $87.96_{\pm0.61}$ | $95.06_{\pm0.30}$ | $57.91_{\pm0.51}$ | $79.90_{\pm0.11}$ | $58.86_{\pm0.79}$ | $79.28_{\pm0.21}$ | $\underline{77.25_{\pm0.10}}$ |
| Energy | $61.34_{\pm4.63}$ | $87.58_{\pm0.46}$ | $41.69_{\pm5.32}$ | $91.21_{\pm0.92}$ | $95.06_{\pm0.30}$ | $55.62_{\pm0.61}$ | $80.91_{\pm0.08}$ | $56.59_{\pm1.38}$ | $79.77_{\pm0.61}$ | $77.25_{\pm0.10}$ |
| Energy+UM | $33.12_{\pm0.47}$ | $90.60_{\pm0.37}$ | $22.95_{\pm1.65}$ | $93.67_{\pm0.68}$ | $92.33_{\pm0.41}$ | $65.86_{\pm2.06}$ | $77.14_{\pm0.62}$ | $\underline{51.90_{\pm2.04}}$ | $81.63_{\pm1.70}$ | $72.21_{\pm0.43}$ |
| Energy+UMAP | $\underline{33.01_{\pm0.06}}$ | $91.00_{\pm0.07}$ | $\underline{21.70_{\pm1.57}}$ | $94.20_{\pm0.36}$ | $95.06_{\pm0.30}$ | $59.71_{\pm0.65}$ | $79.49_{\pm0.23}$ | $52.11_{\pm2.36}$ | $81.62_{\pm1.37}$ | $77.25_{\pm0.10}$ |
| **Energy+PSKD** | $\mathbf{31.67_{\pm0.78}}$ | $\mathbf{91.71_{\pm0.16}}$ | $\mathbf{20.48_{\pm1.30}}$ | $\mathbf{94.56_{\pm0.29}}$ | $\mathbf{95.14_{\pm0.08}}$ | $\underline{54.83_{\pm2.79}}$ | $\mathbf{81.45_{\pm0.44}}$ | $\mathbf{51.56_{\pm2.39}}$ | $\mathbf{82.40_{\pm0.52}}$ | $\mathbf{77.44_{\pm0.09}}$ |
| *Training-time regularization methods (w/o realistic outliers)* | | | | | | | | | | |
| LogitNorm | $29.34_{\pm0.81}$ | $92.33_{\pm0.08}$ | $13.81_{\pm0.20}$ | $96.74_{\pm0.06}$ | $94.30_{\pm0.25}$ | $62.89_{\pm0.57}$ | $78.47_{\pm0.31}$ | $53.61_{\pm3.45}$ | $81.53_{\pm1.26}$ | $76.34_{\pm0.17}$ |
| SNN | $37.21_{\pm0.70}$ | $90.25_{\pm0.09}$ | $26.05_{\pm2.34}$ | $92.49_{\pm0.78}$ | $\mathbf{95.11_{\pm0.13}}$ | $60.32_{\pm1.44}$ | $\underline{80.33_{\pm0.22}}$ | $53.52_{\pm1.77}$ | $82.17_{\pm0.69}$ | $\mathbf{77.56_{\pm0.27}}$ |
| T2FNorm | $\underline{26.47_{\pm0.35}}$ | $92.79_{\pm0.13}$ | $\mathbf{12.75_{\pm0.73}}$ | $96.98_{\pm0.23}$ | $94.69_{\pm0.07}$ | $\underline{58.47_{\pm1.35}}$ | $79.84_{\pm0.40}$ | $51.25_{\pm2.52}$ | $82.73_{\pm1.01}$ | $76.43_{\pm0.13}$ |
| T2FNorm+UM | $27.79_{\pm0.55}$ | $92.74_{\pm0.16}$ | $14.06_{\pm1.31}$ | $96.80_{\pm0.28}$ | $94.02_{\pm0.13}$ | $59.42_{\pm0.74}$ | $79.32_{\pm0.24}$ | $51.09_{\pm0.98}$ | $83.71_{\pm0.37}$ | $75.67_{\pm0.41}$ |
| T2FNorm+UMAP | $26.78_{\pm0.75}$ | $\underline{93.08_{\pm0.28}}$ | $13.11_{\pm0.81}$ | $\mathbf{97.01_{\pm0.19}}$ | $94.69_{\pm0.07}$ | $59.77_{\pm0.03}$ | $79.63_{\pm0.28}$ | $\underline{49.92_{\pm4.16}}$ | $83.81_{\pm1.84}$ | $76.43_{\pm0.13}$ |
| **T2FNorm+PSKD** | $\mathbf{25.72_{\pm0.28}}$ | $\mathbf{93.18_{\pm0.12}}$ | $\underline{13.07_{\pm0.17}}$ | $96.94_{\pm0.11}$ | $95.06_{\pm0.01}$ | $\mathbf{57.26_{\pm0.26}}$ | $\mathbf{80.50_{\pm0.25}}$ | $\mathbf{49.15_{\pm1.67}}$ | $\mathbf{84.13_{\pm0.69}}$ | $\underline{77.14_{\pm0.08}}$ |
| *Training-time regularization methods (w/ realistic outliers)* | | | | | | | | | | |
| MixOE | $51.45_{\pm7.78}$ | $88.73_{\pm0.82}$ | $33.84_{\pm4.77}$ | $91.93_{\pm0.69}$ | $94.55_{\pm0.32}$ | $55.22_{\pm0.49}$ | $80.95_{\pm0.20}$ | $63.88_{\pm2.48}$ | $76.40_{\pm1.44}$ | $75.13_{\pm0.06}$ |
| DOE | $20.39_{\pm0.15}$ | $94.84_{\pm0.07}$ | $15.59_{\pm1.47}$ | $94.67_{\pm0.69}$ | $94.32_{\pm0.19}$ | $37.84_{\pm1.05}$ | $86.61_{\pm0.29}$ | $\mathbf{45.38_{\pm0.52}}$ | $\underline{84.30_{\pm0.81}}$ | $75.69_{\pm0.26}$ |
| OE | $19.84_{\pm0.95}$ | $94.82_{\pm0.21}$ | $13.13_{\pm0.53}$ | $96.00_{\pm0.13}$ | $94.63_{\pm0.26}$ | $30.73_{\pm0.11}$ | $88.30_{\pm0.10}$ | $54.82_{\pm2.79}$ | $81.41_{\pm1.49}$ | $\mathbf{76.84_{\pm0.42}}$ |
| OE+UM | $\mathbf{18.18_{\pm0.97}}$ | $95.04_{\pm0.24}$ | $12.55_{\pm2.63}$ | $96.36_{\pm0.37}$ | $94.51_{\pm0.31}$ | $\underline{30.71_{\pm0.46}}$ | $88.28_{\pm0.23}$ | $53.42_{\pm2.29}$ | $81.92_{\pm1.58}$ | $76.51_{\pm0.37}$ |
| OE+UMAP | $\underline{18.20_{\pm0.51}}$ | $\underline{95.06_{\pm0.14}}$ | $\mathbf{12.12_{\pm2.24}}$ | $96.37_{\pm0.65}$ | $94.63_{\pm0.26}$ | $\mathbf{30.16_{\pm0.08}}$ | $\mathbf{88.44_{\pm0.08}}$ | $54.07_{\pm1.21}$ | $82.42_{\pm0.87}$ | $\mathbf{76.84_{\pm0.42}}$ |
| **OE+PSKD** | $18.55_{\pm0.21}$ | $\mathbf{95.08_{\pm0.12}}$ | $\underline{12.21_{\pm0.53}}$ | $\mathbf{96.58_{\pm0.44}}$ | $\mathbf{94.75_{\pm0.18}}$ | $31.55_{\pm0.57}$ | $88.24_{\pm0.15}$ | $\underline{48.72_{\pm0.74}}$ | $\mathbf{84.60_{\pm0.43}}$ | $\underline{76.99_{\pm0.05}}$ |

**Baselines.** We compare our proposed method with a suite of competitive baselines, including: (1) OOD scoring methods with vanilla training (using cross-entropy loss): MSP (Hendrycks & Gimpel, 2017), ODIN (Liang et al., 2018), and Energy (Liu et al., 2020); (2) training-time regularization methods without realistic outliers: LogitNorm (Wei et al., 2022), T2Fnorm (Regmi et al., 2024), and SNN (Ghosal et al., 2024); and (3) training-time regularization methods with realistic outliers: OE (Hendrycks et al., 2019), MixOE (Zhang et al., 2023a), and DOE (Wang et al., 2023). Moreover, we compare the most relevant methods, UM and UMAP (Zhu et al., 2023), both of which, same as our PSKD, function as plug-in solutions designed to restore the model's intrinsic OOD detection capability.

**Implementation Details.** In line with OpenOOD (Zhang et al., 2023b), we adopt ResNet-18 (He et al., 2016) as the backbone architecture. Models are trained with stochastic gradient descent (SGD) for 100 epochs, using a learning rate of 0.1 with cosine annealing decay schedule (Loshchilov & Hutter, 2017), momentum of 0.9, and weight decay of $5 \times 10^{-4}$. The batch size is set to 128 for CIFAR-10/100 and 256 for ImageNet-200. For validation, 1000 samples from the official test set are used, while the remainder are for testing. The model's self-selection mechanism is applied after the validation step at the end of each epoch. Hyperpa-

rameter selection is performed via a grid search using the ID and OOD validation samples from OpenOOD, and the parameters achieving the best AUROC are used for the final test. We leave more implementation details in Appendix B.

## 5.2. Main Results

In this section, we present a comprehensive evaluation of our proposed PSKD method on both small-scale CIFAR benchmarks, detailed in Table 1, and the more realistic and challenging ImageNet benchmark, summarized in Table 2. The experimental results demonstrate that: (1) On small-scale CIFAR benchmarks, PSKD can consistently improve baseline methods by restoring the model's intrinsic OOD detection capabilities across different OOD scenarios, while also achieving additional gains in ID accuracy. (2) On the large-scale ImageNet benchmark, PSKD continues to exhibit competitive OOD detection performance while achieving the best ID accuracy among the compared methods. (3) Overall, PSKD outperforms the UM/UMAP method across a range of benchmark tests. This demonstrates the efficacy of the self-learning mechanisms, which leverage the continuously evolving self-selection teacher model to effectively push the model beyond its intrinsic OOD detection capability limits.

Table 2: Comparison on large-scale ImageNet benchmark. All values are percentages, and OOD detection results are averaged over multiple OOD test datasets. Detailed results for each OOD dataset are provided in Appendix G.

| Method | Near-OOD | | Far-OOD | | Average | | ID ACC↑ |
|---|---|---|---|---|---|---|---|
| | FPR95↓ | AUROC↑ | FPR95↓ | AUROC↑ | FPR95↓ | AUROC↑ | |
| *OOD scoring methods (vanilla training with cross-entropy)* | | | | | | | |
| MSP | **54.82**±0.35 | **83.34**±0.06 | 35.43±0.38 | 90.13±0.09 | 45.13±0.32 | 86.74±0.07 | 86.37±0.08 |
| ODIN | 66.76±0.26 | 80.27±0.08 | 34.23±1.05 | **91.71**±0.19 | 50.50±0.55 | 85.99±0.13 | 86.37±0.08 |
| Energy | 60.24±0.57 | 82.50±0.05 | 34.86±1.30 | 90.86±0.21 | 47.55±0.76 | 86.68±0.12 | 86.37±0.08 |
| Energy+UM | 60.23±1.13 | 81.79±0.38 | 32.46±1.30 | 91.68±0.37 | 46.34±0.81 | 86.74±0.28 | 85.01±0.31 |
| Energy+UMAP | 60.81±0.84 | 81.08±0.39 | 32.47±0.67 | 91.62±0.29 | 46.64±0.74 | 86.35±0.32 | 86.37±0.08 |
| **Energy+PSKD** | 57.12±0.63 | 82.84±0.32 | **31.64**±0.87 | 91.39±0.16 | **44.38**±0.22 | **87.12**±0.15 | **86.79**±0.25 |
| *Training-time regularization methods (w/o realistic outliers)* | | | | | | | |
| LogitNorm | 57.80±1.22 | 82.21±0.52 | 25.31±0.20 | 93.31±0.15 | 41.56±0.71 | 87.76±0.33 | 86.18±0.60 |
| SNN | 59.85±0.46 | 81.33±0.19 | 28.04±0.64 | 92.28±0.21 | 43.95±0.35 | 86.80±0.16 | 86.56±0.03 |
| T2FNorm | 55.65±0.20 | 82.72±0.05 | 25.25±0.48 | 93.38±0.11 | 40.45±0.33 | 88.06±0.07 | 86.52±0.21 |
| T2FNorm+UM | 55.20±0.71 | **82.89**±0.20 | 26.31±0.91 | 93.21±0.19 | 40.76±0.11 | 88.05±0.06 | 86.31±0.24 |
| T2FNorm+UMAP | 56.14±0.69 | 82.52±0.32 | 26.25±0.30 | 93.02±0.09 | 41.20±0.43 | 87.77±0.19 | 86.52±0.21 |
| **T2FNorm+PSKD** | **55.06**±0.46 | 82.83±0.12 | **24.77**±0.75 | **93.61**±0.17 | **39.91**±0.51 | **88.22**±0.09 | **86.74**±0.09 |
| *Training-time regularization methods (w/ realistic outliers)* | | | | | | | |
| MixOE | 57.95±0.23 | 82.57±0.23 | 40.12±0.66 | 88.39±0.02 | 49.03±0.44 | 85.49±0.12 | 85.73±0.09 |
| DOE | 54.14±0.51 | 83.23±0.60 | 37.60±2.95 | 88.24±1.45 | 45.87±1.72 | 85.73±1.00 | 79.87±3.12 |
| OE | 52.55±0.51 | 84.51±0.21 | 35.18±0.63 | 88.21±0.19 | 43.86±0.49 | 86.36±0.20 | 85.78±0.12 |
| OE+UM | 51.89±0.64 | 84.94±0.17 | 34.76±0.56 | **88.68**±0.07 | 43.33±0.31 | **86.81**±0.08 | 85.71±0.23 |
| OE+UMAP | 52.34±0.22 | 84.56±0.08 | 35.03±0.68 | 88.39±0.18 | 43.68±0.38 | 86.48±0.11 | 85.78±0.12 |
| **OE+PSKD** | **51.88**±0.47 | **84.98**±0.10 | **34.14**±0.33 | 88.61±0.10 | **43.01**±0.12 | 86.79±0.03 | **86.31**±0.11 |

(a) Diff. Schedules for $\lambda$     (b) Diff. Final Weight $\lambda_{\text{final}}$     (c) Diff. Weight $\alpha$     (d) Diff. Temperature $T$

Figure 5: Ablation studies on the hyperparameters: (a) influence of various adjustment strategies for $\lambda$; (b) impact of final weight $\lambda_{\text{final}}$ in cosine annealing schedule; (c) impact of varying the weighting factor $\alpha$ in $R_{\text{OOD}}$; (d) effects of different temperature scaling $T$ for self-distillation.

## 5.3. Ablation Study

**Effect of Adjustment Strategies for $\lambda$.** Figures 5(a) an-
alyze the impact of different $\lambda$-adjustment strategies in
Equation (4), specifically considering the constant strategy
(Const), where $\lambda$ remains fixed; the exponential strategy
(Exp), which leads to rapid initial growth of $\lambda$; the linear
strategy (Linear), where $\lambda$ increases at a constant rate; and
the cosine strategy (Cos) with an initial soft decline de-
scribed in Equation (5). Appendix B.3 provides the details
of various adjustment strategies. The results indicate that
Const and Exp strategies over-rely on the OOD teacher early
in training, preventing the model from effectively learning
ID knowledge, thereby constraining the development of
OOD discrimination capabilities. In contrast, Linear and
Cos strategies better promote ID learning and OOD detec-

tion, with Cos showing the most pronounced improvements.
Figure 5(b) illustrates the impact of varying $\lambda_{\text{final}}$ in Equa-
tion (5) within the Cos strategy. The significant decline
in performance when $\lambda_{\text{final}} = 0$ highlights the importance
of PSKD in OOD learning. Moreover, since the learning
objective of PSKD aims to provide secondary information,
a relatively smaller parameter setting is most beneficial,
whereas excessively large settings may lead to misleading
results and degraded performance.

**Effect of the Weighting Factor $\alpha$.** In Figure 5(c), we evalu-
ate the impact of varying the weighting factor $\alpha$ in Equation
(9), which governs the balance between response-level and
feature-level knowledge in distillation. As we can see, the
performance markedly worsens when feature-level distilla-
tion is excluded (*i.e.*, $\alpha = 0$), underscoring the critical role

Table 3: Impact of varying teacher selection intervals on CIFAR-10 benchmark. OOD results are averaged over near- and far-OOD groups. * denotes the default setting.

| Teacher Selection Interval | FPR95↓ | AUROC↑ | ID ACC↑ |
|---|---|---|---|
| 10 selections per 1 epoch | 26.25±1.46 | 93.11±0.22 | 95.08±0.22 |
| 5 selections per 1 epoch | **25.79**±**2.17** | **93.29**±**0.30** | **95.30**±**0.07** |
| 1 selection per 1 epoch* | 26.08±1.02 | 93.14±0.21 | 95.14±0.08 |
| 1 selection per 5 epochs | 26.86±0.91 | 92.96±0.37 | 95.14±0.17 |

of feature-level OOD discrimination knowledge. Furthermore, a large $\alpha$ places too much emphasis on feature-level distillation, leading to suboptimal performance. Therefore, striking a moderate balance is more beneficial for uncertainty learning, resulting in better performance.

**Effect of Temperature Scaling $T$.** Figure 5(d) examines the impact of temperature scaling $T$ on the uncertainty-embedded target learning in distillation. Compared to the baseline (*i.e.* $T = 1$), higher $T$ facilitates knowledge transfer by softening targets, thereby providing richer discrimination signals for the distillation process and ultimately improving performance. In contrast, excessively high temperatures can overly smooth the teacher signals, resulting in a loss of information. This undermines the intended purpose and degrades overall performance. Thus, a mild temperature provides the best effect for OOD knowledge learning.

**Effect of Teacher Update Frequency.** In Table 3, we examine the effect of teacher model update intervals on training stability and final performance. The results demonstrate that PSKD maintains stable training dynamics across a range of update settings. Notably, appropriately increasing the update frequency can enhance OOD detection performance by providing the model with more frequent guidance. This enables the model to better explore and refine its intrinsic OOD detection capacity. However, such gains come at the cost of increased computational overhead, highlighting a trade-off between performance and efficiency.

### 5.4. Further Analysis

**Impact of Teacher Selection Strategies.** To validate the effectiveness of progressive teacher selection, we design another version of the teacher selection strategy: the stationary strategy (PSKD-S), wherein models are sequentially compared in pairs based on the value of $A(\cdot)$ with Equation (6) throughout the training process to identify the one with the best OOD performance. The model exhibiting the best OOD detection capability is selected as $f_{\theta*}$ and used to guide the retraining of models to learn better OOD discrimination capabilities. The results in Table 4 validate that PSKD performs better than PSKD-S in all OOD detection tasks, which verifies that progressive distillation can further enhance performance by continually refining itself.

Table 4: Teacher selection strategy comparison. PSKD-S refers to the strategy without progressive selection. The best results are in bold, and the second-best are underlined.

| Dataset | Method | FPR95↓ | AUROC↑ | ID ACC↑ |
|---|---|---|---|---|
| CIFAR-10 | Vanilla | 51.52±3.82 | 89.40±0.42 | 95.06±0.30 |
| | +PSKD-S | 28.18±3.29 | 92.82±0.68 | 95.06±0.05 |
| | +PSKD | **26.08**±**1.02** | **93.14**±**0.21** | **95.14**±**0.08** |
| CIFAR-100 | Vanilla | 56.10±0.99 | 80.34±0.34 | 77.25±0.10 |
| | +PSKD-S | 53.40±0.96 | 81.78±0.06 | **77.57**±**0.07** |
| | +PSKD | **53.19**±**1.13** | **81.93**±**0.13** | 77.44±0.09 |
| ImageNet-200 | Vanilla | 47.55±0.76 | 86.68±0.12 | 86.37±0.08 |
| | +PSKD-S | 46.32±0.43 | 86.90±0.16 | **86.86**±**0.13** |
| | +PSKD | **44.38**±**0.22** | **87.12**±**0.15** | 86.79±0.25 |

Table 5: Impact of OOD scoring selection on AUROC based on the CIFAR-10 benchmark.

| Self-selection OOD score | Method | Test-time OOD score | | |
|---|---|---|---|---|
| | | MSP | Energy | KNN |
| / | Vanilla | 89.38±0.18 | 89.40±0.42 | 91.99±0.10 |
| MSP | +PSKD | 91.29±0.43 | 92.79±0.40 | 92.10±0.59 |
| Energy | +PSKD | **91.57**±**0.22** | **93.14**±**0.21** | 92.54±0.10 |
| KNN | +PSKD | 91.24±0.22 | 92.52±0.48 | **92.71**±**0.25** |

**Impact of OOD Scoring Selection.** Table 5 examines the generalization performance of PSKD under different OOD scores selected during self-selection and testing time. The analysis covers various OOD scoring functions, including logit-based methods like the baseline MSP (Hendrycks & Gimpel, 2017), Energy (Liu et al., 2020), and the feature-based KNN (Sun et al., 2022). The results reveal that: (1) Aligning self-selection and test OOD scores (diagonal entries) generally enhances performance. Notably, within the same level, superior self-selection OOD scores better guide teacher self-selection. For example, using Energy scores for self-selection improves MSP test AUROC by 0.28% over matching MSP scores. (2) Even when the score is based on different levels, the model can still drive performance gains through the use of PSKD. For example, using logit-based Energy scores for self-selection and feature-based KNN scores for testing results in a 0.55% AUROC boost over the baseline. (3) Better OOD scores are more critical and lead to improved performance, *i.e.*, the Energy scores for self-selection and testing achieve the best performance.

**Impact of Pseudo-Outlier Generation Strategies.** Table 6 explores the effect of different sources of pseudo-outliers for the teacher model self-selection. Several cost-effective strategies are considered, including image rotation (Rot) to enhance diversity, image distortion (Dist) through resizing followed by restoration, and the addition of Gaussian noise (Gauss) at varying intensities. Specific details are provided in Appendix B.4. The results show that: (1) The

Table 6: The impact of different sources of pseudo-outliers on the CIFAR-10 benchmark. OOD results are averaged over near- and far-OOD groups.

| Source | | | FPR95↓ | AUROC↑ | ID ACC↑ |
|--------|------|-------|--------|--------|---------|
| Rot | Dist | Gauss | | | |
| × | × | × | 51.52±3.82 | 89.40±0.42 | 95.06±0.30 |
| ✓ | ✓ | × | 28.52±1.26 | 92.65±0.13 | 95.07±0.12 |
| ✓ | × | ✓ | 30.88±1.06 | 92.41±0.16 | **95.15±0.16** |
| × | ✓ | ✓ | 27.78±0.71 | 92.87±0.03 | 95.11±0.04 |
| ✓ | ✓ | ✓ | **26.08±1.02** | **93.14±0.21** | 95.14±0.08 |

Dist and Gauss strategies achieved promising results by disrupting image features to construct pseudo-outliers data. (2) Combining the Rot strategy further improves performance by augmenting diversity and expanding the pseudo-outliers distribution. (3) When all considered strategies are combined, the generated pseudo-outliers are the most diverse and informative, leading to the best performance. Overall, performance remains relatively stable across different data quality levels, with high-quality validation sets generally yielding the most significant performance gains.

## 6. Conclusion

In this paper, we explore storing models' intrinsic OOD detection capabilities via self-learning mechanisms, aiming to establish a robust model basic for OOD detection. To address the issue of suboptimal OOD detection performance during the later stages of training, we propose the PSKD framework. This framework progressively refines the model's OOD discrimination capability by leveraging a self-selection teacher model. Extensive experiments demonstrate that PSKD consistently improves OOD detection performance and further enhances the effectiveness of other OOD detection methods.

## Acknowledgements

National Key RD Program of China (2022YFF0712100), NSFC (62276131), Natural Science Foundation of Jiangsu Province of China under Grant (BK20240081), the Fundamental Research Funds for the Central Universities (No.30922010317, No.30923011007), Key Laboratory of Target Cognition and Application Technology (2023-CXPT-LC-005).

## Impact Statement

This paper presents work whose goal is to advance the field of Machine Learning. There are many potential societal consequences of our work, none which we feel must be specifically highlighted here.

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

# Appendix

## A. Pseudo Code of PSKD

The Pseudo code of PSKD is available in Algorithm 1.

---

**Algorithm 1** PKSD: Progressive Self-Knowledge Distillation

---

**Input**: Training dataset $\mathcal{D}_{\text{train}}$, validation dataset $\mathcal{D}_{\text{val}}$, initial model parameters $\theta$
**Output**: Trained student model $f_\theta$

 1: Initialize teacher model: $\theta^* = \theta$
 2: Generate pseudo-outlier dataset $\mathcal{D}_{\text{OOD}}$ from $\mathcal{D}_{\text{val}}$
 3: **while** student model $f_\theta$ has not converged **do**
 4:     Update $f_\theta$ to minimize both $R_{\text{ID}}(f_\theta)$ and $R_{\text{OOD}}(f_\theta, f_{\theta^*})$ with weight factor $\lambda$ in Equation (4) over $\mathcal{D}_{\text{train}}$.
 5:     **if** a predefined number of steps (e.g., one epoch) is completed **then**
 6:         Quantify the OOD detection capability of $f_\theta$ and $f_{\theta^*}$ based on $\mathcal{D}_{\text{val}}$ and $\mathcal{D}_{\text{OOD}}$ following Equation (6): $A(f_\theta)$ and $A(f_{\theta^*})$
 7:         **if** $A(f_{\theta^*}) \leq A(f_\theta)$ **then**
 8:             Update teacher model: $\theta^* = \theta$
 9:         **end if**
10:     **end if**
11:     Adjust weight factor $\lambda$ following Equation (5)
12: **end while**
13: **Return** $f_\theta$

---

## B. Implementation details

### B.1. Software and Hardware

All experiments are conducted using Python 3.8.19 and PyTorch 2.0.1 on a workstation equipped with dual 2.20 GHz CPUs, 384 GB of RAM, and six NVIDIA RTX 4090 GPUs.

### B.2. Validation Strategy

We use the ID and OOD validation samples from OpenOOD for hyperparameters selection of PSKD. We utilize grid search over all possible values of $\lambda_{\text{final}} \times \alpha \times T$ to determine the optimal settings, where we vary $\lambda_{\text{final}} = \{0.01, 0.02, 0.03, 0.05, 0.1\}$, $\alpha = \{1, 5\}$, $T = \{3, 5\}$. The hyperparameter that yields the best AUROC is used for the final test. We adopt the same hyperparameters in the same model in all experiments. For CIFAR benchmarks, the optimal settings are $\lambda_{\text{final}} = 0.01$, $\alpha = 1$ and $T = 3$. For ImageNet-200, $\lambda_{\text{final}} = 0.05$, $\alpha = 1$ and $T = 3$ is optimal.

### B.3. Details of Adjustment Strategies for $\lambda$

We provide details regarding the various $\lambda$ adjustment strategies in Equation (4) below.

Let $\lambda_{\text{final}}$ represent the final weight, and let $E$ denote the total number of training epochs. Four adjustment strategies are considered: constant strategy (Const), exponential adjustment strategy (Exp), linear adjustment strategy (Linear), and cosine annealing adjustment strategy (Cos). The mathematical expressions for these strategies are as follows:

$$\lambda_{\text{Const}} = \lambda_{\text{final}}, \quad \lambda_{\text{Exp}} = \lambda_{\text{final}}\left(1 - \exp\left(-\frac{\beta e}{E}\right)\right), \quad \lambda_{\text{Linear}} = \lambda_{\text{final}} \cdot \frac{e}{E}, \quad \lambda_{\text{Cos}} = \frac{\lambda_{\text{final}}}{2}\left(1 - \cos\left(\frac{e}{E}\pi\right)\right), \quad (10)$$

where $\lambda_{\text{Const}}$ remains fixed at the final weight $\lambda_{\text{final}}$, $\lambda_{\text{Exp}}$ increases following an exponential growth influenced by a parameter $\beta$ (set to 5 in our experiment), $\lambda_{\text{Linear}}$ increases linearly with the epoch number, and $\lambda_{\text{Cos}}$ increases according to a cosine annealing schedule. The visualizations of these adjustment strategies are presented in Figure 6.

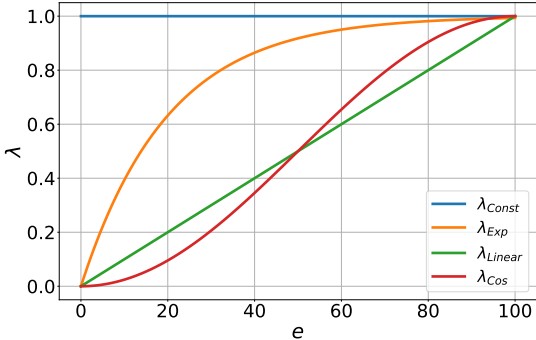

Figure 6: Curves of $\lambda$ under four different adjustment strategies, with the parameters fixed at $\lambda_{\text{final}} = 1$ and $E = 100$.

### B.4. Details of Pseudo-Outlier Generation Strategies

We consider three cost-effective strategies for generating pseudo-outliers: rotation, distortion, and Gaussian noise. The detailed implementation settings are provided below:

**Rotation:** We apply random rotation to enhance the diversity of pseudo-outliers and introduce shifts, with the rotation angle uniformly sampled from $\{90°, 180°, 270°\}$.

**Distortion:** To generate challenging pseudo-outliers that resemble ID samples, we apply a distortion technique. Given an original RGB image of size $(3, h, w)$ and a distortion factor $\gamma \in [0, 1]$, the image is first downsampled to $(3, \lfloor \gamma h \rfloor, \lfloor \gamma w \rfloor)$ using bilinear interpolation. It is then upsampled back to its original dimensions $(3, h, w)$ using bilinear interpolation. This procedure leads to a loss of fine-grained feature information while retaining the essential scene-level content from the original image. For the CIFAR benchmark, the distortion factor is set to 0.16, which corresponds to downsampling an image from its original size of $32 \times 32$ pixels to $5 \times 5$ pixels and then upsampling it back to $32 \times 32$ pixels. For the ImageNet benchmark, the distortion factor is set to 0.05, meaning the original $224 \times 224$ pixel image is downsampled to $11 \times 11$ pixels and then upsampled back to $224 \times 224$ pixels.

**Gaussian Noise:** Noise with varying intensities is added to achieve different levels of random corruption to the original ID images, simulating both near-OOD and far-OOD scenarios. Gaussian noise with a mean of 0 and a variance of $\sigma$ is added to the normalized images, where $\sigma$ takes each of the 10 values from the set $\{0.00, 0.02, 0.04, 0.06, 0.08, 0.10, 0.12, 0.14, 0.16, 0.18\}$. This process expands the dataset by a factor of 10, aiming to address the limited size of the validation set provided by OpenOOD, ultimately facilitating a more accurate identification of the desired OOD teacher model.

## C. Teacher Selection with Realistic Outlier

To ensure a fair comparison, we intentionally avoid using realistic OOD data as auxiliary information in the main paper. To further explore this aspect, Table 7 examines the impact of using realistic outliers for teacher model selection. The results indicate that both realistic and pseudo-outlier data consistently lead to significant performance improvements. Moreover, realistic OOD data can more accurately reflect the actual OOD distribution, thereby enhancing the robustness of teacher model selection and further improving performance.

Table 7: The impact of different sources of OOD data for teacher selection on ImageNet-200 benchmark. Following the settings of the OpenOOD benchmark for the OOD validation set, OpenImage-O is adopted as the realistic OOD data. The OOD results are averaged over near- and far-OOD groups.

| Methods | FPR95↓ | AUROC↑ | ID ACC↑ |
|---|---|---|---|
| Vanilla | 47.55±0.76 | 86.68±0.12 | 86.37±0.08 |
| **+PSKD w/ Pseudo** | 44.38±0.22 | 87.12±0.15 | **86.79±0.25** |
| **+PSKD w/ Realistic** | **43.93±0.27** | **87.64±0.18** | 86.76±0.16 |

## D. Analysis of Computational Overhead

Table 8 presents the additional training overhead introduced by PSKD on both small-scale and large-scale datasets. The results indicate that: (1) the overhead of the teacher selection process accounts for only a small fraction of the total training cost, which is affordable. Notably, for large-scale datasets, the additional overhead of PSKD is even less significant due to the relatively higher ratio of training samples to validation samples. (2) The generation of pseudo-outliers is conducted only once at the beginning of training, and its computational cost is minimal compared to the overall training time.

Table 8: Overhead analysis introduced by PSKD. The training setup follows the OpenOOD benchmark, consisting of a total of 100 epochs for CIFAR-10 (Small scale) and 90 epochs for ImageNet-200 (Large scale). The results are averaged over five independent runs. Appendix B.1 outlines the software and hardware configurations.

| Dataset | One-Epoch Training | One-Time Teacher Selection | Total Training | Pseudo-Outlier Preprocessing |
|---|---|---|---|---|
| CIFAR-10 | 10.50 seconds | 1.18 seconds | 21.96 minutes | 0.96 minutes |
| ImageNet-200 | 252.30 seconds | 8.72 seconds | 397.49 minutes | 1.42 minutes |

## E. Architectural Variants

Table 9 analyzes the robustness of PSKD across different architectures, including the CNN-based ResNet-18 and the transformer-based Vision Transformer (Dosovitskiy et al., 2021) (ViT-B/16). The results indicate that PSKD consistently improves the model's OOD detection performance across different architectures and demonstrates general applicability.

Table 9: The robustness of PSKD across different architectures on the ImageNet-200 benchmark. The Energy score is adopted for OOD scoring and the OOD results are averaged over near- and far-OOD groups.

| Model | Methods | FPR95↓ | AUROC↑ | ID ACC↑ |
|---|---|---|---|---|
| ResNet-18 | Vanilla | $47.55_{\pm0.76}$ | $86.68_{\pm0.12}$ | $86.37_{\pm0.08}$ |
| | **+PSKD** | $\mathbf{44.38_{\pm0.22}}$ | $\mathbf{87.12_{\pm0.15}}$ | $\mathbf{86.79_{\pm0.25}}$ |
| Vit-B/16 | Vanilla | $28.80_{\pm0.41}$ | $93.61_{\pm0.19}$ | $93.90_{\pm0.10}$ |
| | **+PSKD** | $\mathbf{27.79_{\pm0.50}}$ | $\mathbf{93.87_{\pm0.24}}$ | $\mathbf{94.01_{\pm0.06}}$ |

## F. Statistical Tests for Comparison

In this section, we report the statistical test results using AUROC as the OOD detection metric on the CIFAR-10 benchmark. First, we rank the performance of all methods across multiple datasets, with the results recorded in Table 10. We then perform the Friedman test to determine whether there is a significant difference in the average rankings of the methods. The Friedman test yields a test statistic of 25.524 and a p-value of 0.00011 (which is less than the significance level of 0.05), indicating a statistically significant difference among the methods. To further identify specific methods with significant performance differences, we conduct a Nemenyi test, obtaining a Critical Difference (CD) value of 3.078, and present the comparative analysis of ranking differences between our PSKD and other methods in Table 11.

The statistical test results indicate that (1) compared to Energy, Energy+PSKD effectively enhances the model's OOD detection capability and demonstrates a significant performance improvement; and (2) the lack of a significant difference between our PSKD and Unleashing Mask/Unleashing Mask Adopts Pruning (UM/UMAP) can be attributed to our shared goal of restoring the model's intrinsic OOD detection capability. The distinction lies in our method, which utilizes an uncertainty-embedded target to learn valuable atypical samples, whereas UM/UMAP directly discards them, resulting in a loss of ID generalization performance and limiting the model's OOD detection capability.

Table 10: Average ranking results of methods across multiple datasets.

| | MSP | ODIN | Energy | Energy+UM | Energy+UMAP | Energy+PSKD |
|---|---|---|---|---|---|---|
| Average Rank | 4.67 | 5.67 | 4.67 | 2.67 | 1.83 | **1.50** |

Table 11: Ranking differences between our PSKD method and other comparison methods, with an asterisk (*) indicating a significant difference, where the value exceeds the CD value of 3.078.

|  | MSP | ODIN | Energy | Energy+UM | Energy+UMAP |
|---|---|---|---|---|---|
| Ranking Difference | 3.167* | 4.167* | 3.167* | 1.167 | 0.333 |

## G. Fine-grained Results

We present the fine-grained results of our experiments on the CIFAR benchmarks, including six OOD datasets (CIFAR-10/100, TinyImageNet, MNIST, SVHN, Textures, Place365), as well as on ImageNet-200, with five OOD datasets (SSB-hard, NINCO, iNaturalist, Textures, OpenImage-O). The results on the CIFAR-10 benchmark are reported in Table 12 and Table 13, while the results on the CIFAR-100 benchmark are shown in Table 14 and Table 15, and the results on the ImageNet-200 benchmark are presented in Table 16 and Table 17. Our findings include: (1) By equipping with PSKD, the baseline can achieve significant improvements and outperform other methods on most near- and far-OOD datasets. For example, on the CIFAR-10 benchmark, PSKD can reduce FPR95 by 31.26% on CIFAR-100 (near-OOD) and 28.26% on Textures (far-OOD). (2) In both w/o and w/ realistic outliers scenarios, the results consistently demonstrate that as a plugin, PSKD effectively excavates the model's intrinsic capabilities and pushes their performance further. For instance, on the CIFAR-100 benchmark, PSKD reduced the FPR95 of OE on SVHN by 17.17%. (3) In addition to the OOD detection performance improvement, PSKD can also improve ID accuracy by reasonably learning atypical samples. PSKD consistently achieved improvements of 0.08%, 0.19%, and 0.42% in ID test accuracy on CIFAR-10, CIFAR-100, and ImageNet-200, respectively. This contrasts with UM/UMAP, which directly forgets valuable atypical samples. Our proposed PSKD leverages the uncertainty-embedded targets to learn from both typical and atypical samples, which achieves a dual success in both OOD detection performance and main task performance.

**Analysis of Performance Disparities.** The performance improvement of PSKD is inherently dependent on the model's intrinsic OOD capability relative to the training data. Compared to the small-scale CIFAR dataset, the large-scale ImageNet dataset encompasses a larger and more complex semantic space. Under the same ResNet-18 architecture used in our paper, models trained on the more challenging ImageNet dataset tend to learn a more crowded feature space, increased class overlap, and unreliable decision boundaries. These factors contribute to the model's relatively weak intrinsic OOD detection capability (Huang & Li, 2021), limiting PSKD's potential to restore the model's intrinsic OOD detection capability. Consequently, the performance improvements on ImageNet are less pronounced, while substantial gains are observed on the simpler CIFAR-10 benchmark. Although the improvements vary across datasets of different scales, PSKD consistently enhances the model's OOD detection performance, proving its effectiveness.

Table 12: Fine-grained results (AUROC↑) on the CIFAR-10 benchmark.

| Method | Near-OOD | | | Far-OOD | | | | | ID ACC |
|---|---|---|---|---|---|---|---|---|---|
| | CIFAR-100 | TinyImageNet | Average | MNIST | SVHN | Textures | Places365 | Average | |
| *OOD scoring methods (vanilla training with cross-entropy)* | | | | | | | | | |
| MSP | 87.19±0.33 | 88.87±0.19 | 88.03±0.25 | 92.63±1.57 | 91.46±0.40 | 89.89±0.71 | 88.92±0.47 | 90.73±0.43 | 95.06±0.30 |
| ODIN | 82.18±1.87 | 83.55±1.84 | 82.87±1.85 | 95.24±1.96 | 84.58±0.77 | 86.94±2.26 | 85.07±1.24 | 87.96±0.61 | 95.06±0.30 |
| Energy | 86.36±0.58 | 88.80±0.36 | 87.58±0.46 | 94.32±2.53 | 91.79±0.98 | 89.47±0.70 | 89.25±0.78 | 91.21±0.92 | 95.06±0.30 |
| Energy+UM | 89.08±0.33 | 92.12±0.44 | 90.60±0.37 | 96.86±0.87 | 92.29±2.86 | 92.10±0.53 | 93.41±0.59 | 93.67±0.68 | 92.33±0.41 |
| Energy+UMAP | 89.67±0.18 | 92.33±0.09 | 91.00±0.07 | **97.70±0.53** | 94.30±1.69 | 91.23±0.32 | **93.58±0.48** | 94.20±0.36 | 95.06±0.30 |
| **Energy+PSKD** | **90.62±0.16** | **92.80±0.16** | **91.71±0.16** | 97.20±0.30 | **94.98±1.57** | **93.38±0.37** | 92.69±0.40 | **94.56±0.29** | **95.14±0.08** |
| *Training-time regularization methods (w/o realistic outliers)* | | | | | | | | | |
| LogitNorm | 90.95±0.22 | 93.70±0.06 | 92.33±0.08 | 99.14±0.45 | 98.25±0.41 | 94.77±0.43 | **94.79±0.16** | 96.74±0.06 | 94.30±0.25 |
| SNN | 89.32±0.09 | 91.18±0.09 | 90.25±0.09 | 93.53±1.04 | 91.98±1.76 | 92.94±0.34 | 91.49±0.23 | 92.49±0.78 | **95.11±0.13** |
| T2FNorm | 91.56±0.10 | 94.02±0.16 | 92.79±0.13 | 99.28±0.27 | **98.81±0.22** | 95.44±0.77 | 94.40±0.32 | 96.98±0.23 | 94.69±0.07 |
| T2FNorm+UM | 91.21±0.20 | 94.28±0.12 | 92.74±0.16 | 99.19±0.21 | 98.12±0.08 | 95.10±0.64 | **94.79±0.30** | 96.80±0.28 | 94.02±0.13 |
| T2FNorm+UMAP | 91.69±0.28 | **94.47±0.28** | 93.08±0.28 | 99.37±0.22 | 98.63±0.35 | 95.30±0.13 | 94.74±0.27 | **97.01±0.19** | 94.69±0.07 |
| **T2FNorm+PSKD** | **91.95±0.11** | 94.41±0.15 | **93.18±0.12** | 99.23±0.19 | 98.50±0.34 | **95.47±0.71** | 94.54±0.11 | 96.94±0.11 | 95.06±0.01 |
| *Training-time regularization methods (w/ realistic outliers)* | | | | | | | | | |
| MixOE | 87.47±0.97 | 90.00±0.73 | 88.73±0.82 | 91.66±2.21 | 93.82±1.27 | 91.84±0.51 | 90.38±0.55 | 91.93±0.69 | 94.55±0.32 |
| DOE | 92.63±0.21 | 97.04±0.27 | 94.84±0.07 | 85.16±2.20 | 99.20±0.09 | 97.77±0.31 | 96.57±0.18 | 94.67±0.69 | 94.32±0.19 |
| OE | 90.54±0.53 | 99.11±0.34 | 94.82±0.21 | 90.22±1.31 | 99.60±0.14 | 97.58±0.27 | 96.58±0.70 | 96.00±0.13 | 94.63±0.26 |
| OE+UM | 90.65±0.66 | 99.42±0.34 | 95.04±0.24 | 92.85±2.16 | 99.72±0.17 | 97.03±1.14 | 95.83±2.16 | 96.36±0.37 | 94.51±0.31 |
| OE+UMAP | 90.58±0.60 | **99.55±0.32** | 95.06±0.14 | 92.97±3.57 | **99.78±0.08** | 97.34±1.14 | 95.37±2.27 | 96.37±0.65 | 94.63±0.26 |
| **OE+PSKD** | **90.66±0.26** | 99.50±0.15 | **95.08±0.12** | 91.80±2.19 | 99.73±0.11 | **97.89±0.11** | **96.91±0.37** | 96.58±0.44 | **94.75±0.18** |

Table 13: Fine-grained results (FPR95↓) on the CIFAR-10 benchmark.

| Method | Near-OOD | | | Far-OOD | | | | | ID ACC |
|---|---|---|---|---|---|---|---|---|---|
| | CIFAR-100 | TinyImageNet | Average | MNIST | SVHN | Textures | Places365 | Average | |
| *OOD scoring methods (vanilla training with cross-entropy)* | | | | | | | | | |
| MSP | 53.08±4.86 | 43.27±3.00 | 48.17±3.92 | 23.64±5.81 | 25.82±1.64 | 34.96±4.64 | 42.47±3.81 | 31.72±1.84 | 95.06±0.30 |
| ODIN | 77.00±5.74 | 75.38±6.42 | 76.19±6.08 | 23.83±12.34 | 68.61±0.52 | 67.70±11.06 | 70.36±6.96 | 57.62±4.24 | 95.06±0.30 |
| Energy | 66.60±4.46 | 56.08±4.83 | 61.34±4.63 | 24.99±12.93 | 35.12±6.11 | 51.82±6.11 | 54.85±6.52 | 41.69±5.32 | 95.06±0.30 |
| Energy+UM | 38.33±0.27 | **27.92±0.94** | 33.12±0.47 | 11.58±3.50 | 26.30±7.48 | 27.24±2.78 | 26.67±1.99 | 22.95±1.65 | 92.33±0.41 |
| Energy+UMAP | 37.49±0.42 | 28.54±0.44 | 33.01±0.06 | **8.59±1.81** | 20.11±6.72 | | **26.49±1.88** | 21.70±1.57 | 95.06±0.30 |
| **Energy+PSKD** | **35.34±0.94** | 28.01±0.73 | **31.67±0.78** | 11.41±0.50 | **17.60±4.85** | 23.56±1.48 | 29.34±2.41 | **20.48±1.30** | 95.14±0.08 |
| *Training-time regularization methods (w/o realistic outliers)* | | | | | | | | | |
| LogitNorm | 34.37±1.30 | 24.30±0.54 | 29.34±0.81 | 3.93±1.99 | 8.33±1.78 | 21.94±0.85 | **21.04±0.71** | 13.81±0.20 | 94.30±0.25 |
| SNN | 41.00±0.76 | 33.43±0.82 | 37.21±0.70 | 22.92±3.86 | 25.33±3.64 | 24.34±1.77 | 31.60±1.04 | 26.05±2.34 | **95.11±0.13** |
| T2FNorm | 30.60±0.45 | 22.33±0.37 | 26.47±0.35 | 3.50±1.33 | **5.72±0.66** | 19.49±2.58 | 22.27±1.28 | **12.75±0.73** | 94.69±0.07 |
| T2FNorm+UM | 32.60±0.80 | 22.98±0.34 | 27.79±0.55 | 3.63±0.96 | 8.39±0.95 | 21.81±2.79 | 22.39±0.91 | 14.06±1.31 | 94.02±0.13 |
| T2FNorm+UMAP | 31.61±0.72 | 21.95±0.79 | 26.78±0.75 | **2.95±1.12** | 6.87±2.01 | 20.95±0.17 | 21.67±0.72 | 13.11±0.81 | 94.69±0.07 |
| **T2FNorm+PSKD** | **29.72±0.51** | **21.72±0.52** | **25.72±0.28** | 3.67±0.93 | 7.03±1.01 | **19.48±2.51** | 22.11±0.33 | 13.07±0.17 | 95.06±0.01 |
| *Training-time regularization methods (w/ realistic outliers)* | | | | | | | | | |
| MixOE | 58.29±8.25 | 44.62±7.57 | 51.45±7.78 | 38.28±13.40 | 20.36±3.99 | 33.19±4.28 | 43.54±4.95 | 33.84±4.77 | 94.55±0.32 |
| DOE | **28.27±0.84** | 12.50±0.95 | 20.39±0.15 | 35.70±3.33 | 2.55±0.78 | **10.35±1.50** | **13.77±0.47** | 15.59±1.47 | 94.32±0.19 |
| OE | 36.71±2.06 | 2.97±1.17 | 19.84±0.95 | 24.67±2.55 | 1.25±0.36 | 12.07±2.14 | 14.53±2.80 | 13.13±0.53 | 94.63±0.26 |
| OE+UM | 34.34±2.07 | 2.01±1.18 | **18.18±0.97** | 19.20±1.74 | 0.99±0.60 | 13.87±5.65 | 16.12±6.99 | 12.55±2.63 | 94.51±0.31 |
| OE+UMAP | 34.78±1.68 | 1.61±1.36 | 18.20±0.51 | **16.86±4.94** | **0.77±0.18** | 12.70±4.77 | 18.16±7.02 | **12.12±2.24** | 94.63±0.26 |
| **OE+PSKD** | 35.58±0.71 | **1.51±0.48** | 18.55±0.21 | 22.37±3.83 | 0.78±0.11 | 11.68±0.67 | 14.00±1.54 | 12.21±0.53 | **94.75±0.18** |

Table 14: Fine-grained results (AUROC↑) on the CIFAR-100 benchmark.

| Method | Near-OOD | | | Far-OOD | | | | | ID ACC |
|---|---|---|---|---|---|---|---|---|---|
| | CIFAR-10 | TinyImageNet | Average | MNIST | SVHN | Textures | Places365 | Average | |
| *OOD scoring methods (vanilla training with cross-entropy)* | | | | | | | | | |
| MSP | 78.47±0.07 | 82.07±0.17 | 80.27±0.11 | 76.08±1.86 | 78.42±0.89 | 77.32±0.71 | 79.22±0.29 | 77.76±0.44 | 77.25±0.10 |
| ODIN | 78.18±0.14 | 81.63±0.08 | 79.90±0.11 | 83.79±1.31 | 74.54±0.76 | **79.33±1.08** | 79.45±0.26 | 79.28±0.21 | 77.25±0.10 |
| Energy | 79.05±0.11 | 82.76±0.08 | 80.91±0.08 | 79.18±1.37 | 82.03±1.74 | 78.35±0.83 | 79.52±0.23 | 79.77±0.61 | 77.25±0.10 |
| Energy+UM | 72.86±1.03 | 81.43±0.58 | 77.14±0.62 | **86.39±2.97** | 86.00±3.61 | 76.58±1.03 | 77.54±1.44 | 81.63±1.70 | 72.21±0.43 |
| Energy+UMAP | 76.96±0.46 | 82.01±0.15 | 79.49±0.23 | 84.90±4.24 | 85.25±3.16 | 78.16±1.36 | 78.17±0.42 | 81.62±1.37 | 77.25±0.10 |
| **Energy+PSKD** | **79.16±0.93** | **83.74±0.18** | **81.45±0.44** | 84.19±0.78 | **86.20±1.32** | 78.64±0.85 | 80.59±0.67 | **82.40±0.52** | 77.44±0.09 |
| *Training-time regularization methods (w/o realistic outliers)* | | | | | | | | | |
| LogitNorm | 74.57±0.39 | 82.37±0.24 | 78.47±0.31 | 90.69±1.38 | 82.80±4.57 | 72.37±0.67 | 80.25±0.61 | 81.53±1.26 | 76.34±0.17 |
| SNN | **76.99±0.25** | 83.68±0.20 | 80.33±0.22 | 82.90±1.21 | 83.06±2.44 | 82.98±0.37 | 79.74±0.27 | 82.17±0.69 | **77.56±0.27** |
| T2FNorm | 76.09±0.81 | 83.59±0.02 | 79.84±0.40 | 86.22±2.29 | 86.04±1.04 | 77.32±1.63 | 81.35±0.33 | 82.73±1.01 | 76.43±0.13 |
| T2FNorm+UM | 75.21±0.34 | 83.43±0.23 | 79.32±0.24 | **90.89±1.97** | 86.22±1.95 | 77.63±0.82 | 80.10±0.67 | 83.71±0.37 | 75.67±0.41 |
| T2FNorm+UMAP | 75.69±0.27 | 83.56±0.37 | 79.63±0.28 | 90.58±0.68 | 85.67±5.89 | 77.72±1.34 | 81.28±0.47 | 83.81±1.84 | 76.43±0.13 |
| **T2FNorm+PSKD** | 76.95±0.33 | **84.04±0.18** | 80.50±0.25 | 87.17±3.19 | **89.94±1.41** | 77.80±0.44 | 81.60±0.31 | **84.13±0.69** | 77.14±0.08 |
| *Training-time regularization methods (w/ realistic outliers)* | | | | | | | | | |
| MixOE | **78.17±0.29** | 83.73±0.12 | 80.95±0.20 | 76.06±5.52 | 72.28±0.81 | 77.34±0.91 | 79.92±0.30 | 76.40±1.44 | 75.13±0.06 |
| DOE | 75.47±0.55 | 97.76±0.28 | 86.61±0.29 | 72.54±4.38 | **95.73±0.42** | **86.34±1.28** | **82.58±0.91** | 84.30±0.81 | 75.69±0.26 |
| OE | 76.70±0.19 | 99.89±0.02 | 88.30±0.10 | 80.68±5.82 | 84.37±1.34 | 82.18±0.68 | 78.39±0.41 | 81.41±1.49 | 76.84±0.42 |
| OE+UM | 76.67±0.48 | 99.89±0.03 | 88.28±0.23 | 81.44±3.24 | 84.32±2.00 | 83.16±1.35 | 78.76±0.58 | 81.92±1.58 | 76.51±0.37 |
| OE+UMAP | 76.95±0.16 | **99.92±0.00** | **88.44±0.08** | **85.27±1.68** | 83.92±3.16 | 82.14±0.68 | 78.35±0.35 | 82.42±0.87 | 76.84±0.42 |
| **OE+PSKD** | 76.56±0.31 | 99.91±0.02 | 88.24±0.15 | 82.64±0.85 | 92.38±1.67 | 83.75±0.80 | 79.64±0.17 | **84.60±0.43** | 76.99±0.05 |

Table 15: Fine-grained results (FPR95↓) on the CIFAR-100 benchmark.

| Method | Near-OOD | | | Far-OOD | | | | | ID ACC |
|---|---|---|---|---|---|---|---|---|---|
| | CIFAR-10 | TinyImageNet | Average | MNIST | SVHN | Textures | Places365 | Average | |
| *OOD scoring methods (vanilla training with cross-entropy)* | | | | | | | | | |
| MSP | **58.91±0.93** | 50.70±0.34 | **54.80±0.33** | 57.23±4.68 | 59.07±2.53 | **61.88±1.28** | 56.62±0.87 | 58.70±1.06 | 77.25±0.10 |
| ODIN | 60.64±0.56 | 55.19±0.57 | 57.91±0.51 | 45.94±3.29 | 67.41±3.88 | 62.37±2.96 | 59.71±0.92 | 58.86±0.79 | 77.25±0.10 |
| Energy | 59.21±0.75 | 52.03±0.50 | 55.62±0.61 | 52.62±3.83 | 53.62±3.14 | 62.35±2.06 | 57.75±0.86 | 56.59±1.38 | 77.25±0.10 |
| Energy+UM | 74.91±4.09 | 56.81±1.04 | 65.86±2.06 | **36.56±5.81** | **39.72±7.86** | 68.30±2.69 | 63.00±3.26 | 51.90±2.04 | 72.21±0.43 |
| Energy+UMAP | 64.73±0.97 | 54.70±0.33 | 59.71±0.65 | 42.80±7.56 | 40.03±6.52 | 65.02±4.09 | 60.61±1.61 | 52.11±2.36 | 77.25±0.10 |
| **Energy+PSKD** | 60.73±4.96 | **48.93±0.63** | 54.83±2.79 | 42.91±2.95 | 44.76±4.73 | 63.23±2.45 | **55.34±0.60** | **51.56±2.39** | **77.44±0.09** |
| *Training-time regularization methods (w/o realistic outliers)* | | | | | | | | | |
| LogitNorm | 73.88±1.21 | 51.89±0.10 | 62.89±0.57 | 34.12±8.32 | 47.52±8.02 | 77.38±2.99 | 55.44±1.45 | 53.61±3.45 | 76.34±0.17 |
| SNN | 71.56±2.17 | **49.08±0.73** | 60.32±1.44 | 46.31±1.23 | 54.11±6.62 | **54.14±1.30** | 59.51±0.91 | 53.52±1.77 | 77.56±0.27 |
| T2FNorm | 67.07±1.90 | 49.88±0.85 | 58.47±1.35 | 39.39±5.38 | 44.29±3.14 | 66.82±4.61 | 54.50±0.52 | 51.25±2.52 | 76.43±0.13 |
| T2FNorm+UM | 69.07±0.79 | 49.77±0.68 | 59.42±0.74 | 34.64±6.54 | 46.56±6.07 | 67.14±2.49 | 56.00±1.26 | 51.09±0.98 | 75.67±0.41 |
| T2FNorm+UMAP | 69.30±0.66 | 50.25±0.60 | 59.77±0.03 | **32.48±2.22** | 46.25±15.16 | 66.88±2.55 | 54.06±0.60 | 49.92±4.16 | 76.43±0.13 |
| **T2FNorm+PSKD** | **65.18±0.77** | 49.33±0.47 | **57.26±0.26** | 41.69±10.12 | **35.43±5.75** | 65.46±1.46 | 54.00±0.72 | **49.15±1.67** | 77.14±0.08 |
| *Training-time regularization methods (w/ realistic outliers)* | | | | | | | | | |
| MixOE | 61.12±1.08 | 49.32±0.36 | 55.22±0.49 | 59.49±7.74 | 73.09±4.00 | 66.04±0.98 | 56.93±0.78 | 63.88±2.48 | 75.13±0.06 |
| DOE | 63.85±0.50 | 11.83±1.60 | 37.84±1.05 | 57.97±4.42 | **20.27±1.01** | 50.29±2.73 | 52.97±1.56 | 45.38±0.52 | 75.69±0.26 |
| OE | 61.26±0.22 | 0.21±0.01 | 30.73±0.11 | 53.31±9.91 | 51.84±3.45 | 55.83±1.82 | 58.30±0.72 | 54.82±2.79 | 76.84±0.42 |
| OE+UM | 61.15±0.95 | 0.27±0.03 | 30.71±0.46 | 50.43±4.09 | 49.87±3.58 | 54.79±2.52 | 58.59±1.37 | 53.42±2.29 | 76.51±0.37 |
| OE+UMAP | **60.15±0.12** | **0.18±0.04** | **30.16±0.08** | **49.14±1.83** | 52.76±5.43 | 55.77±1.47 | 58.62±0.71 | 54.07±1.21 | 76.84±0.42 |
| **OE+PSKD** | 62.91±1.13 | 0.19±0.10 | 31.55±0.57 | 50.30±4.22 | 34.67±3.87 | 52.94±2.22 | 56.95±0.40 | **48.72±0.74** | **76.99±0.05** |

Table 16: Fine-grained results (AUROC↑) on the ImageNet-200 benchmark.

| Method | Near-OOD | | | Far-OOD | | | | ID ACC |
|---|---|---|---|---|---|---|---|---|
| | SSB-hard | NINCO | Average | iNaturalist | Textures | OpenImage-O | Average | |
| *OOD scoring methods (vanilla training with cross-entropy)* | | | | | | | | |
| MSP | **80.38±0.03** | **86.29±0.11** | **83.34±0.06** | 92.80±0.25 | 88.36±0.13 | 89.24±0.02 | 90.13±0.09 | 86.37±0.08 |
| ODIN | 77.19±0.06 | 83.34±0.12 | 80.27±0.08 | **94.37±0.41** | 90.65±0.20 | **90.11±0.15** | **91.71±0.19** | 86.37±0.08 |
| Energy | 79.83±0.02 | 85.17±0.11 | 82.50±0.05 | 92.55±0.50 | 90.79±0.16 | 89.23±0.26 | 90.86±0.21 | 86.37±0.08 |
| Energy+UM | 79.15±0.46 | 84.43±0.29 | 81.79±0.38 | 93.75±0.56 | 92.11±0.41 | 89.18±0.46 | 91.68±0.37 | 85.01±0.31 |
| Energy+UMAP | 77.94±0.42 | 84.22±0.66 | 81.08±0.39 | 93.25±0.69 | **93.19±0.16** | 88.42±0.35 | 91.62±0.29 | 86.37±0.08 |
| **Energy+PSKD** | 79.87±0.29 | 85.81±0.42 | 82.84±0.32 | 92.97±0.19 | 91.80±0.31 | 89.40±0.26 | 91.39±0.16 | **86.79±0.25** |
| *Training-time regularization methods (w/o realistic outliers)* | | | | | | | | |
| LogitNorm | 78.18±0.59 | 86.24±0.46 | 82.21±0.52 | 96.40±0.23 | 92.36±0.24 | 91.17±0.47 | 93.31±0.15 | 86.18±0.60 |
| SNN | 77.47±0.24 | 85.18±0.31 | 81.33±0.19 | 92.82±0.36 | **94.55±0.10** | 89.45±0.19 | 92.28±0.21 | 86.56±0.03 |
| T2FNorm | 78.82±0.08 | 86.63±0.04 | 82.72±0.05 | 96.72±0.21 | 91.97±0.28 | 91.47±0.21 | 93.38±0.11 | 86.52±0.21 |
| T2FNorm+UM | **79.25±0.31** | 86.53±0.11 | **82.89±0.20** | 96.52±0.25 | 91.80±0.26 | 91.32±0.26 | 93.21±0.19 | 86.31±0.24 |
| T2FNorm+UMAP | 78.77±0.24 | 86.27±0.41 | 82.52±0.32 | 96.36±0.10 | 91.65±0.27 | 91.06±0.08 | 93.02±0.09 | 86.52±0.21 |
| **T2FNorm+PSKD** | 78.82±0.11 | **86.83±0.20** | 82.83±0.12 | **96.79±0.10** | 92.28±0.34 | **91.75±0.08** | **93.61±0.17** | **86.74±0.09** |
| *Training-time regularization methods (w/ realistic outliers)* | | | | | | | | |
| MixOE | 80.15±0.21 | 84.99±0.28 | 82.57±0.23 | **90.94±0.17** | 87.02±0.22 | 87.22±0.04 | 88.39±0.02 | 85.73±0.09 |
| DOE | 80.64±0.71 | 85.81±0.54 | 83.23±0.60 | 90.64±1.21 | 87.04±1.74 | 87.05±1.41 | 88.24±1.45 | 79.87±3.12 |
| OE | 82.14±0.17 | 86.89±0.25 | 84.51±0.21 | 89.08±0.18 | 87.33±0.22 | 88.22±0.19 | 88.21±0.19 | 85.78±0.12 |
| OE+UM | **82.57±0.31** | 87.32±0.18 | 84.94±0.17 | 89.96±0.05 | 87.50±0.21 | **88.59±0.07** | **88.68±0.07** | 85.71±0.23 |
| OE+UMAP | 82.30±0.16 | 86.83±0.08 | 84.56±0.08 | 89.37±0.28 | 87.44±0.06 | 88.37±0.20 | 88.39±0.18 | 85.78±0.12 |
| **OE+PSKD** | 82.56±0.12 | **87.39±0.13** | **84.98±0.10** | 89.51±0.10 | **87.85±0.19** | 88.47±0.07 | 88.61±0.10 | **86.31±0.11** |

Table 17: Fine-grained results (FPR95↓) on the ImageNet-200 benchmark.

| Method | Near-OOD | | | Far-OOD | | | | ID ACC |
|---|---|---|---|---|---|---|---|---|
| | SSB-hard | NINCO | Average | iNaturalist | Textures | OpenImage-O | Average | |
| *OOD scoring methods (vanilla training with cross-entropy)* | | | | | | | | |
| MSP | **66.00±0.10** | **43.65±0.75** | **54.82±0.35** | 26.48±0.73 | 44.58±0.68 | **35.23±0.18** | 35.43±0.38 | 86.37±0.08 |
| ODIN | 73.51±0.38 | 60.00±0.80 | 66.76±0.26 | **22.39±1.87** | 42.99±1.56 | 37.30±0.59 | 34.23±1.05 | 86.37±0.08 |
| Energy | 69.77±0.32 | 50.70±0.89 | 60.24±0.57 | 26.41±2.29 | 41.43±1.85 | 36.74±1.14 | 34.86±1.30 | 86.37±0.08 |
| Energy+UM | 68.91±1.49 | 51.55±0.82 | 60.23±1.13 | 24.69±1.40 | 35.46±1.94 | 37.23±1.30 | 32.46±1.30 | 85.01±0.31 |
| Energy+UMAP | 70.33±0.65 | 51.28±1.55 | 60.81±0.84 | 26.23±1.76 | **31.71±0.75** | 39.46±1.02 | 32.47±0.67 | 86.37±0.08 |
| **Energy+PSKD** | 67.88±0.33 | 46.36±1.34 | 57.12±0.63 | 24.11±0.93 | 35.05±1.30 | 35.77±1.19 | **31.64±0.87** | **86.79±0.25** |
| *Training-time regularization methods (w/o realistic outliers)* | | | | | | | | |
| LogitNorm | 68.07±0.77 | 47.52±1.71 | 57.80±1.22 | 15.10±1.26 | **31.21±1.79** | 29.62±1.08 | 25.31±0.20 | 86.18±0.60 |
| SNN | 73.11±0.93 | 46.60±0.28 | 59.85±0.46 | 25.66±0.76 | 24.81±0.66 | 33.64±0.60 | 28.04±0.64 | 86.56±0.03 |
| T2FNorm | 66.41±0.26 | 44.88±0.54 | 55.65±0.20 | 13.99±0.68 | 32.39±1.41 | 29.38±0.60 | 25.25±0.48 | 86.52±0.21 |
| T2FNorm+UM | **65.27±0.88** | 45.13±0.65 | 55.20±0.71 | 14.73±1.12 | 34.32±1.87 | 29.89±0.39 | 26.31±0.91 | 86.31±0.24 |
| T2FNorm+UMAP | 66.62±0.55 | 45.66±0.93 | 56.14±0.69 | 15.07±0.14 | 33.58±0.69 | 30.11±0.26 | 26.25±0.30 | 86.52±0.21 |
| **T2FNorm+PSKD** | 65.97±0.30 | **44.14±0.63** | **55.06±0.46** | 13.54±0.08 | 31.92±1.78 | **28.84±0.42** | 24.77±0.75 | **86.74±0.09** |
| *Training-time regularization methods (w/ realistic outliers)* | | | | | | | | |
| MixOE | 67.94±0.43 | 47.95±0.63 | 57.95±0.23 | 29.97±0.20 | 50.23±1.21 | 40.17±0.67 | 40.12±0.66 | 85.73±0.09 |
| DOE | 63.70±0.04 | 44.59±0.99 | 54.14±0.51 | 29.13±3.75 | 44.87±1.93 | 38.81±3.35 | 37.60±2.95 | 79.87±3.12 |
| OE | 64.20±0.45 | 40.90±0.66 | 52.55±0.51 | 28.93±0.58 | 41.82±1.09 | 34.78±0.24 | 35.18±0.63 | 85.78±0.12 |
| OE+UM | **63.07±0.83** | 40.71±0.65 | 51.89±0.64 | **27.59±0.31** | 42.19±1.64 | 34.51±0.36 | 34.76±0.56 | 85.71±0.23 |
| OE+UMAP | 63.48±0.10 | 41.20±0.35 | 52.34±0.22 | 28.81±0.64 | 41.62±0.85 | 34.66±0.59 | 35.03±0.68 | 85.78±0.12 |
| **OE+PSKD** | 63.57±0.79 | **40.19±0.15** | **51.88±0.47** | 27.97±0.38 | **40.70±0.88** | 33.76±0.29 | 34.14±0.33 | **86.31±0.11** |

