# OpenReview forum: "Strengthen Out-of-Distribution Detection Capability with Progressive Self-Knowledge Distillation"
_ICML.cc/2025/Conference — ICML 2025 poster_

### Official Review · Reviewer_JJ2e · 2025-02-26

**Overall Recommendation:** 4

**Summary:**

To address the issue of suboptimal OOD detection performance during the later stages of training, this paper proposes Progressive Self-Knowledge Distillation (PSKD) framework. PSKD strengthens the OOD detection capability by leveraging self-provided uncertainty embedded targets. PSKD is orthogonal to most existing methods and thus can be used to further enhance the effectiveness of other OOD detection methods.

## update after rebuttal
After reading the authors' responses, I decide to keep my original score.

**Claims And Evidence:**

Yes, all claims made in the submission are supported by clear and convincing evidence.

**Essential References Not Discussed:**

No, there are no essential references not discussed.

**Experimental Designs Or Analyses:**

Yes, the experimental design and analyses in this paper are solid, with a clear comparison to state-of-the-art methods.

**Methods And Evaluation Criteria:**

Yes, both proposed methods and evaluation criteria make sense for the problem and application.

**Other Comments Or Suggestions:**

Please give the full names for all abbreviations when they occur for the first time. For example, UM/UMAP. I have not any other comments or suggestions.

**Other Strengths And Weaknesses:**

Strengths:
1) To address the issue of suboptimal OOD detection performance during the later stages of training, this paper proposes Progressive Self-Knowledge Distillation (PSKD) framework. PSKD strengthens the OOD detection capability by leveraging self-provided uncertaintyembedded targets.
2) The authors conduct extensive experiments to verify the effectiveness of the proposed PSKD on both small-scale CIFAR and large-scale ImageNet, covering near-OOD scenarios with semantic shifts and far-OOD scenarios with further obvious covariance shifts.
3) The paper is generally a good paper with a clear central idea. The organization of the paper is quite good and it is easy to follow the topic and the proposed algorithms.

Weaknesses:
1) With regard to the comparison results, statistical tests are needed in the comparison results. The detailed description about statistical tests for comparisons of multiple algorithms on multiple datasets can be found from the following paper: Statistical comparisons of classifiers over multiple data sets.
2) In the current version, the authors use the words "approach", "method", “technique”, “framework” and "strategy" a little casually. I know, it is very difficult to distinguish these words thoroughly. At least, the author should try to unify the use of these words in a paper.
3) In the paper, the authors use AUROC to denote the area under the receiver operating characteristic curve. Why? To my knowledge, the area under the receiver operating characteristic curve is widely denoted as AUC. It is not necessary to change the well-known abbreviation.

**Questions For Authors:**

1) With regard to the comparison results, statistical tests are needed in the comparison results. The detailed description about statistical tests for comparisons of multiple algorithms on multiple datasets can be found from the following paper: Statistical comparisons of classifiers over multiple data sets.
2) In the current version, the authors use the words "approach", "method", “technique”, “framework” and "strategy" a little casually. I know, it is very difficult to distinguish these words thoroughly. At least, the author should try to unify the use of these words in a paper.
3) In the paper, the authors use AUROC to denote the area under the receiver operating characteristic curve. Why? To my knowledge, the area under the receiver operating characteristic curve is widely denoted as AUC. It is not necessary to change the well-known abbreviation.

**Relation To Broader Scientific Literature:**

Out-of-distribution (OOD) detection aims to ensure AI system reliability by rejecting inputs outside the training distribution. This paper proposes Progressive Self-Knowledge Distillation (PSKD) framework to improve OOD detection performance and further enhance the effectiveness of other OOD detection methods.

**Theoretical Claims:**

Yes, both proofs for the theoretical claims are correct.

---

> ### Author Rebuttal · Authors · 2025-03-31
>
> Thank you for your positive assessment and helpful feedback.
>
> **Comment 1. Statistical Tests for Comparison:**
> Thank you for your valuable suggestion. The discussion on statistical tests for result comparison will be included in the final version. Here, we report the statistical test results using AUC as the OOD detection metric on the CIFAR-10 benchmark. First, we rank the performance of all methods across multiple datasets, with the results recorded in Table 1. We then perform the Friedman test to determine whether there is a significant difference in the average rankings of the methods. The Friedman test yields a test statistic of 25.524 and a p-value of 0.00011 (which is less than the significance level of 0.05), indicating a statistically significant difference among the methods. To further identify specific methods with significant performance differences, we conduct a Nemenyi test, obtaining a Critical Difference (CD) value of 3.078, and present the comparative analysis of ranking differences between our PSKD and other methods in Table 2.
>
> The statistical test results indicate that (1) compared to Energy, Energy+PSKD effectively enhances the model's OOD detection capability and demonstrates a significant performance improvement; and (2) the lack of a significant difference between our PSKD and Unleashing Mask/Unleashing Mask Adopts Pruning (UM/UMAP) can be attributed to our shared goal of restoring the model’s intrinsic OOD detection capability. The distinction lies in our method, which utilizes an uncertainty-embedded target to learn valuable atypical samples, whereas UM/UMAP directly discards them, resulting in a loss of ID generalization performance and limiting the model's OOD detection capability.
>
> Table 1. Average ranking results of methods across multiple datasets.
> ||MSP|ODIN|Energy|Energy+UM|Energy+UMAP|Energy+PSKD(Ours)|
> |-|-|-|-|-|-|-|
> |Average Rank|4.67|5.67|4.67|2.67|1.83|**1.50**|
>
> Table 2. Ranking differences between our PSKD method and other comparison methods, with an asterisk (*) indicating a significant difference, where the value exceeds the CD value of 3.078.
> ||MSP|ODIN|Energy|Energy+UM|Energy+UMAP|
> |-|-|-|-|-|-|
> |Ranking Difference|3.167*|4.167*|3.167*|1.167|0.333|
>
> **Comment 2. Clarity and Consistency:**
> Thank you for pointing out these mistakes, and we will revise the manuscript accordingly. (1) The full name will be provided upon the first occurrence of any abbreviation to enhance clarity. (2) Special attention will be given to word choice to ensure consistency throughout the manuscript.
>
> **Comment 3. Why Use AUROC Instead of AUC?**
> Thank you for your correction. The abbreviation "AUROC" (Area Under the Receiver Operating Characteristic Curve) used in our paper follows the OpenOOD benchmark [1]. Indeed, the more commonly used abbreviation is "AUC". We will make this revision in the final version.
>
> **Reference:**
>
> [1] Yang, et al. OpenOOD: Benchmarking Generalized Out-of-Distribution Detection. NeurIPS 2022.

---

### Official Review · Reviewer_MqyY · 2025-03-09

**Overall Recommendation:** 4

**Summary:**

This paper proposes Progressive Self-Knowledge Distillation (PSKD), a framework that leverages self-distillation and dynamic teacher selection to enhance a model’s intrinsic OOD detection capability. PSKD uses pseudo-outlier samples generated through rotation, distortion, and Gaussian noise to iteratively refine the student model’s OOD performance by distilling knowledge from a dynamically updated teacher model. Extensive experiments demonstrate the superiority of the PSKD, including (1) main comparison with SOTA OOD detection baselines, (2) ablation study for adjustment strategy, weighting factor, and temperature scaling, (3) other deep analysis. The paper is well-written and easy to follow.

**Claims And Evidence:**

Yes. The key claims made in the paper are well-supported by relevant methodological and experimental evidence. The authors thoroughly explain their approach, ensuring that each assertion is backed by rigorous theoretical analysis and empirical validation. The experimental results are comprehensive and demonstrate the effectiveness of the proposed method across different scenarios, further strengthening the credibility of the findings.

**Essential References Not Discussed:**

Key references are cited in this paper. The author provides a comprehensive and well-structured discussion of related work.

**Experimental Designs Or Analyses:**

Yes. I have evaluated the rationality and effectiveness of the experimental design in the paper. The experiments are well-designed, and the author has provided the relevant code. Based on the experimental setup and results, I think relevant results are reasonable and convincing.

**Methods And Evaluation Criteria:**

Yes. The method proposed by the author addresses OOD problems and effectively tackles related challenges in practical applications.

**Other Comments Or Suggestions:**

N/A

**Other Strengths And Weaknesses:**

Strengths:
1. Novelty: The authors design a novel strategy to adaptively select a self-teacher model. PSKD integrates self-distillation with dynamic teacher selection, addressing the “forgetting” problem in traditional regularization methods.
2. Simplicity and Efficiency: Avoids complex architectures or additional data, relying solely on pseudo-outliers and parameter tuning.
3. Comprehensive Experiments: The authors provide comprehensive experiments to verify the superiority of the proposed method. Furthermore, the authors also report the ablation study, deeply analysis experimental results to further reveal the justifiability and effectiveness.
4. Clearly Writing: This paper is well-written and easy to follow. The logic is clear, and the structure is well-organized, allowing readers to grasp the main ideas efficiently.
Weaknesses:
1. Training Instability: Frequent teacher updating and adaptive $\lambda$ adjustments might be leading to training instability in practice.
2. Computation Overhead: The PSKD requires frequent teacher updates and pseudo-outlier generation, potentially increasing training time.

**Questions For Authors:**

1. Please analyze the training instability problem.
2. Please analyze the training computation overhead problem.

**Relation To Broader Scientific Literature:**

OOD detection is a crucial area in machine learning. This paper addresses the limitation of directly forgetting atypical samples in previous studies.

**Theoretical Claims:**

N/A. The paper does not include any theoretical claims.

---

> ### Author Rebuttal · Authors · 2025-03-31
>
> Thank you for your positive assessment and helpful feedback.
>
> **Comment 1. Analysis of Training Stability:**
> Tables 6-11 in Appendix C of the original paper report the performance of PSKD across multiple independent training runs on various benchmarks. The results show that PSKD exhibits a standard deviation comparable to that of Vanilla training, suggesting that the introduction of PSKD has a negligible effect on the stability of the training process. Below, we provide an analysis of the effects of teacher update frequency and λ adjustment on training stability:
>
> - **Teacher Update Frequency:** Table 1 examines the effect of teacher model update intervals on training stability. The results indicate that PSKD is robust to update frequency, with frequent updates having a limited effect on training stability. Moreover, appropriately increasing the update frequency may help improve OOD detection performance by providing more opportunities for the model to explore its optimal intrinsic OOD detection capability.
>
> - **λ adjustments:** The purpose of adjusting λ is to minimize the influence of self-distillation in the early stages of training. This helps prevent the large bias introduced by the teacher model, which may not have been adequately trained, from destabilizing the training process. As training progresses, λ gradually increases to emphasize the objective of learning the uncertainty-embedded targets. As analyzed in Figure 5(c) of the original paper, failing to adjust λ dynamically may compromise training stability due to excessive interference from an undertrained teacher model.
>
>
> Table 1. The impact of varying teacher selection intervals on training stability on CIFAR-10 benchmark. The OOD results are averaged over near- and far-OOD groups on the CIFAR-10 benchmark.
> |Teacher Selection Interval|FPR95↓|AUROC↑|ID ACC↑
> |-|-|-|-|
> |10 selections per 1 epoch|26.25 ± 1.46|93.11 ± 0.22|95.08 ± 0.22|
> |5 selections per 1 epoch|**25.79 ± 2.17**|**93.29 ± 0.30**|**95.30 ± 0.07**|
> |1 selection per 1 epoch (default)|26.08 ± 1.02|93.14 ± 0.21|95.14 ± 0.08|
> |1 selection per 5 epochs|26.86 ± 0.91|92.96 ± 0.37|95.14 ± 0.17|
> |Single selection throughout training (PSKD-S)|28.18 ± 3.29|92.82 ± 0.68|95.06 ± 0.05|
>
>
> **Comment 2. Analysis of Computational Overhead:**
> Table 2 presents the additional training overhead introduced by PSKD on both small-scale and large-scale datasets. The results indicate that: (1) the overhead of the teacher selection process accounts for only a small fraction of the total training cost, which is affordable. Notably, for large-scale datasets, the additional overhead of PSKD is even less significant due to the relatively higher ratio of training samples to validation samples. (2) The generation of pseudo-outliers is conducted only once at the beginning of training, and its computational cost is minimal compared to the overall training time.
>
> Table 2. Overhead analysis introduced by PSKD. The training setup follows the OpenOOD benchmark [1], consisting of a total of 100 epochs for CIFAR-10 and 90 epochs for ImageNet-200. The results are averaged over five independent runs. Software and hardware configurations remain consistent with those detailed in Appendix B.1 of our paper.
> |Dataset|One-Epoch Training Cost|One-Time Teacher Selection Cost|Total Training Cost|Pseudo-Outlier Preprocessing|
> |-|-|-|-|-|
> |CIFAR-10 (Small scale)|10.50 seconds|1.18 seconds|21.96 minutes|0.96 minutes|
> |ImageNet-200 (Large scale)|252.30 seconds|8.72 seconds|397.49 minutes|1.42 minutes|
>
> **Reference:**
>
> [1] Yang, et al. OpenOOD: Benchmarking Generalized Out-of-Distribution Detection. NeurIPS 2022.

---

### Official Review · Reviewer_nFr5 · 2025-03-12

**Overall Recommendation:** 4

**Summary:**

This paper concerns the out-of-distribution (OOD) detection task. Recent work shows that memorizing atypical samples during later stages of training can hurt OOD detection, while strategies for forgetting them show promising improvements. However, directly forgetting atypical samples sacrifices ID generalization and limits the model's OOD detection capability. To address this issue, this paper proposes Progressive Self-Knowledge Distillation (PSKD), which strengthens the OOD detection capability by leveraging self-provided uncertainty-embedded targets. Specifically, PSKD adaptively selects a self-teacher model from the training history using pseudo-outliers, facilitating the learning of uncertainty knowledge via multi-level distillation applied to features and responses. Moreover, PSKD is orthogonal to most existing methods and can be integrated as a plugin to collaborate with them. Extensive experiments including main comparison and ablation study verify the effectiveness of PSKD.

**Claims And Evidence:**

Clear. The main claims have been supported from methodology and comprehensive experiments.

**Essential References Not Discussed:**

No, the author provides extensive references on OOD detection.

**Experimental Designs Or Analyses:**

The experimental designs or analyses make sense and are validate

**Methods And Evaluation Criteria:**

Make sense.

**Other Comments Or Suggestions:**

It is suggested that the author add the analysis of PSKD performance differences on different datasets in the experimental part, especially those data sets with insignificant performance improvement.

**Other Strengths And Weaknesses:**

Strengths:
Novelty: The proposed PSKD framework is innovative and can effectively solve the problem of overfitting atypical samples in the late training period, thereby improving the OOD detection capability.
Effectiveness: The experimental design was comprehensive, covering multiple datasets and baseline methods, verifying the effectiveness and generality of PSKD. The authors have provided code to enhance the reproducibility of the paper.
Writing and Paper Organization: The writing of the paper is clear, with well-structured arguments and a logical flow that enhances readability and comprehension. And the experimental results makes the paper more convincing.
Cons:
Performance: Although PSKD performed well on multiple datasets, performance improvements on some specific datasets were not significant and may require further analysis for reasons.
Discussion: There is a lack of in-depth discussion on why PSKD can effectively improve OOD detection capability.

**Questions For Authors:**

The performance of PSKD varies greatly on different data sets. Are there some data set characteristics that affect the performance of PSKD? Can you further analyze the reasons for these differences?

**Relation To Broader Scientific Literature:**

The key contributions of the paper are related to the OOD detection, which is an interesting research issue.

**Theoretical Claims:**

This paper does not include theoretical analysis.

---

> ### Author Rebuttal · Authors · 2025-03-31
>
> Thank you for your positive assessment and helpful feedback.
>
> **Comment 1. Analysis of Performance Disparities:**
> The performance improvement of PSKD is inherently dependent on the model's intrinsic OOD capability relative to the training data. Compared to the small-scale CIFAR dataset, the large-scale ImageNet dataset encompasses a larger and more complex semantic space. Under the same ResNet-18 architecture used in our paper, models trained on the more challenging ImageNet dataset tend to learn a more crowded feature space, increased class overlap, and unreliable decision boundaries. These factors contribute to the model's relatively weak intrinsic OOD detection capability [1], limiting PSKD's potential to restore the model's intrinsic OOD detection capability. Consequently, the performance improvements on ImageNet are less pronounced, while substantial gains are observed on the simpler CIFAR-10 benchmark. Although the improvements vary across datasets of different scales, PSKD consistently enhances the model's OOD detection performance, demonstrating its effectiveness.
>
> **Comment 2. Insight Justification of PSKD:**
> Our motivation stems from the observation that the model's OOD detection performs optimally at an intermediate stage of training rather than in the final well-trained state. We attribute this to the issue of label assignment in traditional supervised learning, where samples with varying levels of uncertainty are assigned a uniform, absolutely confident learning target (i.e., one-hot labels). This forces the model to learn overly confident predictions for atypical samples, leading to an overestimation of certainty. Consequently, the model becomes prone to overconfidence regarding OOD data, which hurts OOD detection.
>
> To address this problem, our PSKD leverages the potential within the model's own learning process to generate soft targets that inherently capture sample-level uncertainty. Specifically, PSKD alleviates overfitting to atypical samples by learning from the uncertainty-embedded targets provided by the self-selected teacher model, thereby effectively enhancing the model's ability to perceive uncertainty. As a result, PSKD can effectively learn from atypical samples to restore the model's intrinsic OOD detection capability. Further empirical analysis and interpretative insights regarding PSKD can be found in Section 4.4 of the original paper.
>
> **Reference:**
>
> [1] Huang, et al. MOS: Towards Scaling Out-of-distribution Detection for Large Semantic Space. CVPR 2021.

---

### Official Review · Reviewer_ioAG · 2025-03-12

**Overall Recommendation:** 3

**Summary:**

This paper proposes Progressive Self-Knowledge Distillation (PSKD), a framework to enhance out-of-distribution (OOD) detection by leveraging uncertainty-embedded targets from a self-selected teacher model. The authors argue that models tend to memorize atypical samples during later training stages, harming OOD detection. PSKD dynamically selects a teacher model from training history using pseudo-outliers and applies multi-level distillation (feature and response) to learn uncertainty knowledge. Experiments on CIFAR and ImageNet benchmarks demonstrate improved OOD detection and in-distribution (ID) classification.

**Claims And Evidence:**

Yes, the claims made in the submission are supported by clear and convincing evidence including methodology, comprehensive experiments.

**Essential References Not Discussed:**

No, the references of the paper are comprehensive.

**Experimental Designs Or Analyses:**

Yes, the soundness/validity of experimental designs and analysis is checked.

**Methods And Evaluation Criteria:**

Yes, PSKD makes sense for the OOD detection problem.

**Other Comments Or Suggestions:**

How does PSKD handle distribution shifts in high-dimensional or non-image data (e.g., text or audio)?

**Other Strengths And Weaknesses:**

Strengths:
1. Dynamic Teacher Selection: The AUROC-based criterion (Eq. 6) for selecting teachers is intuitive and aligns with OOD detection objectives.
2. Strong Empirical Validation: Comprehensive experiments across near- and far-OOD scenarios demonstrate PSKD’s superiority over existing methods, e.g., reducing FPR95 by 31.26% on CIFAR-100.
3. Practical Plug-and-Play Design: PSKD’s compatibility with existing OOD scoring methods (e.g., Energy, MSP) and training paradigms (e.g., OE) enhances its applicability.
4. Reproducibility: Open-sourced code and hyperparameter details (Appendix B) ensure transparency.
Weaknesses:
1. Pseudo-Outlier Dependency: Performance hinges on artificially generated pseudo-outliers (Table 5); robustness to realistic OOD data remains understudied.
2. Overlooked Architectural Variants: Evaluations are limited to ResNet-18; performance on transformers or attention-based models is unexplored.

**Questions For Authors:**

Could the teacher selection mechanism be sensitive to the quality of pseudo-outliers? How robust is PSKD to noisy validation data?
What is the computational overhead of the teacher selection process, especially for large-scale training?

**Relation To Broader Scientific Literature:**

Yes, the method proposed in the paper is highly inspiring to the field of OOD detection.

**Theoretical Claims:**

This paper does not include theoretical analysis.

---

> ### Author Rebuttal · Authors · 2025-03-31
>
> Thank you for your positive assessment and helpful feedback.
>
> **Comment 1. Teacher Selection with Realistic Outlier:**
> To ensure a fair comparison, we intentionally avoid using realistic OOD data as auxiliary information following the setting of [1]. To further explore this, we introduce realistic OOD data for teacher selection to evaluate the performance of PSKD, with the results recorded in Table 1. The results indicate that: (1) both realistic and pseudo-outlier data consistently lead to significant performance improvements; (2) realistic OOD data more accurately reflect the actual OOD distribution, thereby enhancing the robustness of teacher model selection and further improving performance.
>
> Table 1. The impact of different sources of OOD data for teacher selection on ImageNet-200 benchmark. Following the settings of the OpenOOD benchmark [1] for the OOD validation set, OpenImage-O is adopted as the realistic OOD data. The OOD results are averaged over near- and far-OOD groups.
> |Methods|FPR95↓|AUROC↑|ID ACC↑
> |-|-|-|-|
> |Vanilla|47.55 ± 0.76|86.68 ± 0.12|86.37 ± 0.08|
> |PSKD w/ Pseudo|44.38 ± 0.22|87.12 ± 0.15|**86.79 ± 0.25**|
> |PSKD w/ Realistic|**43.93 ± 0.27**|**87.64 ± 0.18**|86.76 ± 0.16|
>
> **Comment 2. Architectural Variants:** Table 2 analyzes the robustness of PSKD across different architectures, including the CNN-based ResNet-18 and the transformer-based Vision Transformer [2] (ViT-B/16). The results indicate that PSKD consistently improves the model's OOD detection performance across different architectures and demonstrates general applicability.
>
> Table 2. The robustness of PSKD across different architectures on the ImageNet-200 benchmark. The Energy score is adopted for OOD scoring and the OOD results are averaged over near- and far-OOD groups.
> |Model|Methods|FPR95↓|AUROC↑|ID ACC↑|
> |-|-|-|-|-|
> |ResNet-18|Vanilla|47.55 ± 0.76|86.68 ± 0.12|86.37 ± 0.08|
> |ResNet-18|PSKD|**44.38 ± 0.22**|**87.12 ± 0.15**|**86.79 ± 0.25**|
> |Vit-B/16|Vanilla|28.80 ± 0.41|93.61 ± 0.19|93.90 ± 0.10|
> |Vit-B/16|PSKD|**27.79 ± 0.50**|**93.87 ± 0.24**|**94.01 ± 0.06**|
>
> **Comment 3. Handling of Non-Image Data:**
> A general and feasible way for handling non-image data (e.g., text, audio) is to induce distribution shifts by adding noise after converting the data into embedding vectors. To verify the effectiveness of this strategy, we conduct an OOD detection task in an audio classification setting based on the well-known Kinetics-Sound dataset [3]. The results presented in Table 3 provide empirical evidence that our PSKD is also suitable for other types of non-image data.
>
> Table 3. The applicability of PSKD to audio data, with the Energy score used for OOD scoring.
> |Methods|FPR95↓|AUROC↑|ID ACC↑|
> |-|-|-|-|
> |Vanilla|76.95 ± 1.64|69.67 ± 0.59|68.06 ± 0.60|
> |PSKD|**72.93 ± 1.84**|**71.04 ± 0.48**|**68.33 ± 0.37**|
>
> **Comment 4. Sensitivity to the Quality of the Validation Set:**
> The ablation analysis in Table 5 of the original paper examines the impact of pseudo-outlier quality on OOD detection performance. The results reveal the following: (1) even when using simple pseudo-outlier construction strategies (such as Gaussian noise or distortions), there is a notable improvement in OOD detection performance; (2) increasing the diversity of pseudo-outliers (e.g., by incorporating rotations, distortions, and noise) provides a marginal but further performance boost. Additionally, as shown in Table 1, high-quality, realistic OOD data results in a slight yet consistent improvement in performance. Overall, performance remains relatively stable across different data quality levels, with high-quality validation sets generally yielding the most significant performance gains.
>
> **Comment 5. Computational Overhead of Teacher Selection:**
> Table 4 records the overhead of teacher selection in PSKD on both small-scale and large-scale datasets. The results show that the overhead of the teacher selection process accounts for only a small fraction of the total training cost, which is affordable. Notably, for large-scale datasets, the additional overhead of PSKD is even less significant due to the relatively higher ratio of training samples to validation samples.
>
> Table 4. Analysis of teacher selection overhead in PSKD. The results are averaged over five independent runs. Software and hardware configurations remain consistent with those detailed in Appendix B.1 of our paper.
> |Dataset|One-Epoch Training Cost|One-Time Teacher Selection Cost|
> |-|-|-|
> |CIFAR-10 (Small scale)|10.50 seconds|1.18 seconds|
> |ImageNet-200 (Large scale)|252.30 seconds|8.72 seconds|
>
> **References:**
>
> [1] Yang, et al. OpenOOD: Benchmarking Generalized Out-of-Distribution Detection. NeurIPS 2022.
>
> [2] Dosovitskiy, et al. An Image is Worth 16x16 Words: Transformers for Image Recognition at Scale. ICLR 2021.
>
> [3] Arandjelovic, et al. Look, listen and learn. ICCV 2017.

---

### Decision · Program_Chairs · 2025-05-01

**Decision:**

Accept (poster)

**Comment:**

This paper presents a novel and practical PSKD framework that enhances OOD detection while preserving ID generalization. The approach is novel and practical, with good potential for integration into existing methods. I recommend accept.